# The revised FLORIDyn model: Implementation of heterogeneous flow and the Gaussian wake

Marcus Becker[1], Bastian Ritter[2], Bart Doekemeijer[3], Daan van der Hoek[1], Ulrich Konigorski[2], Dries Allaerts[4], and Jan-Willem van Wingerden[1]

[1]Delft Center for Systems and Control, Delft University of Technology, Mekelweg 2, 2628 CD Delft, Netherlands
[2]Control Systems & Mechatronics Lab, Technische Universität Darmstadt, Landgraf Georg Str. 4, 64283 Darmstadt, Germany
[3]National Renewable Energy Laboratory, Golden, CO, 80401, USA
[4]Faculty of Aerospace Engineering, Delft University of Technology, Kluyverweg 1, 2629 HS Delft, Netherlands

**Correspondence:** Marcus Becker (marcus.becker@tudelft.nl)

**Abstract.**

In this paper, a new version of the FLOw Redirection and Induction Dynamics (FLORIDyn) model is presented. The new model uses the three-dimensional parametric Gaussian FLORIS model and can provide dynamic wind farm simulations at a low computational cost under heterogeneous and changing wind conditions.

Both FLORIS and FLORIDyn are parametric models which can be used to simulate wind farms, evaluate controller performance and can serve as a control-oriented model. One central element in which they differ is in their representation of flow dynamics: FLORIS neglects these and provides a computationally very cheap approximation of the mean wind farm flow. FLORIDyn defines a framework which utilizes this low computational cost of FLORIS to simulate basic wake dynamics. This is achieved by creating so called Observation Points (OPs) at each time step at the rotor plane which inherit the turbine state.

In this work, we develop the initial FLORIDyn framework further considering multiple aspects. The underlying FLORIS wake model is replaced by a Gaussian wake model. The distribution and characteristics of the OPs are adapted to account for the new parametric model, but also to take complex flow conditions into account. To achieve this, a mathematical approach is developed to combine the parametric model and the changing, heterogeneous world conditions and link them with each OP. We also present a computational lightweight wind field model to allow for a simulation environment in which heterogeneous
flow conditions are possible.

FLORIDyn is compared to SOWFA simulations in three- and nine-turbine cases under static and changing environmental conditions. The results show a good agreement with the timing of the impact of upstream state changes on downstream turbines. They also show a good agreement in terms of how wakes are displaced by wind direction changes and when the resulting velocity deficit is experienced by downstream turbines. A good fit of the mean generated power is ensured by the underlying
FLORIS model. In the three turbine case, FLORIDyn simulates $4\,\mathrm{s}$ simulation time in $24.49\,\mathrm{ms}$ computational time. The resulting new FLORIDyn model proves to be a computationally attractive and capable tool for model based dynamic wind farm control.

# 1 Introduction

In recent years, the topic of wind farm control has gained traction as renewable energies become more and more relevant for the current and future energy mix. To maximize the power generated by a wind farm is not a trivial task as the turbine-to-turbine interaction is characterized by the complex flow, large delay times and an ever-changing environment. In order to describe the wind field, parametric steady-state approximations have been developed. These describe the mean behavior of the flow with parametrized analytical expressions rather than differential equations. A first approach was presented by Jensen (1983) which motivated years later the development of more refined steady-state models, such as the Zone FLORIS model (Gebraad et al., 2014). With these low computational cost and easy-to-implement wake descriptions it is possible to develop a model-based control algorithm. These control strategies have managed to improve the power generated in high fidelity simulations e.g. (Gebraad et al., 2014) and in field experiments (Fleming et al., 2017). The success of parametric steady-state models opens up the question of whether it is possible to overcome one of their great shortcomings: The lack of dynamics. A low computational cost dynamic wake description can be used to more accurately describe the wake behavior on smaller time scales, during turbine state changes and during environmental changes. This could lead to more sophisticated control approaches and wind farm analyses methods.

There have been efforts to implement parametric models in a dynamic manner, some of which are described here. For a more in-depth discussion of the current state of the art, the interested reader is referred to the review by Kheirabadi and Nagamune (2019) and more recently, Andersson et al. (2021). In the current literature, we have identified two major trails of publications, which will be briefly discussed below.

The first research trail begins with the *Aeolus SimWindFarm Toolbox* (Grunnet et al., 2010) which is publicly available. The toolbox uses the *Jensen model* (Jensen, 1983), coupled with a dynamic description of the centerline and a wind field grid. The centerline would imitate the wake meandering effect based on passive tracers, traveling with the synthetically generated turbulent wind speed. A number of limitations have been imposed for this toolbox: The mean wind speed and direction are constant, the flow field is calculated in 2D, and the turbines operate with fixed yaw angles. The toolbox has enabled the work of Poushpas and Leithead (2014), who used the *Frandsen multiple wake model* (Frandsen et al., 2006) and added a description of turbine dynamics to estimate fatigue loads. The model is then used to perform induction control based on look-up tables of the thrust and power coefficients with the goal to redistribute loads. This work has later inspired the *dynamic wind farm simulator*, introduced in Bossanyi (2018). The model adds wake steering to the fatigue load estimation and induction control capabilities. To model the effect of yawing the turbine, the deflection formulation of Jiménez et al. (2010) is used. Based on data from the in-house code *Bladed*, the author formulates the effect of yaw misalignment on the power coefficient by a polynomial expression based on the blade pitch. The wind field is represented by low- and a high-frequency wind speed variations. The low-frequency variations are correlated across the wind farm and cause wake meandering and advection. The high-frequency part is uncorrelated between the turbines and is superimposed with the wake deficits. Lastly, the wake model is switched to the *Ainslie model* (Ainslie, 1988).

A second trail of publications can be found starting with Shapiro et al. (2017), where the authors use the previously mentioned Jensen model and extend it to incorporate the impact of time-varying extraction of kinetic energy of turbines due to induction control. Assuming a constant wind direction and wind speed, the authors derive a linear approximation of the wake advection velocity based on the laws of momentum conservation and mass conservation. The result is a one-dimensional partial differential equation to describe the dynamic wake behavior. The model neglects possible changes of the wake expansion due to a changing thrust coefficient and also does not incorporate yaw angle changes. In Shapiro et al. (2018), the authors extend their model to also take the effects of yawing into account. Most recently, this approach inspired the development of the *Floating Offshore Wind Farm Simulator*, published in Kheirabadi and Nagamune (2021). The authors extend the momentum conservation equations to incorporate time-varying free stream wind velocity effects. Additionally, they couple the model to a dynamic description of floating platforms, restricted by mooring lines. The authors closely follow Bastankhah and Porté-Agel (2016) to derive a parametric Gaussian velocity shape for their model.

Alongside the two discussed trails of publications, the *Dynamic Wake Meandering* (DWM) Model was developed. The DWM model, first presented by Larsen et al. (2008) and later calibrated and refined by Madsen et al. (2010), proposes an approach much closer to established CFD methods. The model follows a pseudo-Lagrangian approach and creates turbulence boxes around the wake deficit which is created by the turbine. These boxes are then subject to a synthetic turbulent wind field, which allows the modeling of the wake meandering effect. The DWM model puts a focus on load estimation next to the power generated and simulates the turbine by coupling a CFD actuator disc model with an aeroelastic model. Compared to the other mentioned models, the DWM model presents a synergy of CFD methods with engineering approaches.

Another early attempt to derive a dynamic model from a parametric steady-state model was published by Gebraad and van Wingerden (2014) who utilized the just published *FLORIS* model (FLOw Redirection and Induction in Steady-state, Gebraad et al. (2014)) and created the *FLORIDyn* model (FLOw Redirection and Induction Dynamics). *FLORIDyn* creates so called Observation Points (OPs) at the rotor plane which travel downstream at hub height with the effective wind speed. Their path follows the zone boundaries described by the *FLORIS* model. The wake deficit and shape depend on the yaw angle and the induction factor. Changes in these variables travel with the OPs and cause a delayed effect at downstream turbines. The authors derive a state-space representation of the model behavior and validate it in a six-turbine simulation against the high fidelity large eddy simulation environment *SOWFA* (National Renewable Energy Laboratory, 2020). The state-space representation is then used to implement a Kalman Filter for flow field estimation (Gebraad et al., 2015). The model does have short comings: due to the two-dimensional flow, shear and veer effects can not be captured, the simulations only work in one wind direction and they do not capture turbulent effects. Furthermore, due to the way the OPs travel, parts of the wake can overlap and can create a faulty wake representation.

In this paper, we aim to overcome these issues and bring the *FLORIDyn* approach into a form where it can incorporate heterogeneous and changing flow conditions, wind shear and added turbulence levels. To achieve these changes, we rework the framework to use a *Gaussian FLORIS* model (Bastankhah and Porté-Agel, 2016). This requires a new formulation of the OP behavior. Due to these changes, the wakes can also incorporate locally different and changing flow conditions, such as wind speed, direction and ambient turbulence intensity. To drive the model, a concept of a wind field model is presented as well. The

framework is then compared to the simulation environment *SOWFA* in three and nine turbine cases. Furthermore, in order to allow for collaboration and extension, the code is published in its entirety (Becker, 2022a). The resulting *Gaussian FLORIDyn* model is a capable, open-source alternative to the few other existing in-house parametric dynamic models, developed for wind farm control purposes.

The remainder of this paper is organized as follows: Section 2 discusses the relevant characteristics of the former *FLORIDyn* framework and how it is adapted. The simulation results are presented in Section 3, which also discusses the computational performance. Section 4 concludes the paper and gives recommendations for future work.

## 2    A new parametric dynamic wind farm model

In this Section, the new Gaussian FLORIDyn model is introduced. To prevent confusion, we will refer to the models of Gebraad
et al. as the Zone FLORIS model (Gebraad et al., 2014) and the Zone FLORIDyn model (Gebraad and van Wingerden, 2014). The Gaussian model by Bastankhah and Porté-Agel (2016) will be referred to as Gaussian FLORIS model.

As the new Gaussian FLORIDyn model is building upon previous work, Sections 2.1 and 2.2 briefly introduce the terminology and properties of the underlying Gaussian FLORIS model and the Zone FLORIDyn framework. The novel Gaussian FLORIDyn model makes changes to the Zone FLORIDyn framework. These are discussed in Section 2.3. Section 2.4 describes
how heterogeneous environmental conditions are taken into account. To get the power coefficient ($C_\text{P}$) and the thrust coefficient ($C_\text{T}$) values closer to the validation platform SOWFA, a lookup table was generated (Section 2.5). Lastly, a basic wind field model is given in Section 2.6. It is built to provide the heterogeneous field conditions to evaluate the FLORIDyn model.

In the wake coordinate system, $\mathcal{K}_1$, $x_1$ describes the downwind direction, $y_1$ the horizontal crosswind direction and $z_1$ the vertical crosswind direction (Fig. 1). In this coordinate frame, the rotor center is always located at $[0, 0, 0]^\top$. This coordinate
system is not to be confused with the longitudinal ($x_0$), latitudinal ($y_0$) and vertical ($z_0$) world coordinate system $\mathcal{K}_0$.

### 2.1    The Gaussian FLORIS model

The core of the used Gaussian FLORIS model is based on the work of Bastankhah and Porté-Agel (2016). This work describes a parametric, three-dimensional wake with a Gaussian shaped wind speed recovery. As it has been applied and described in previous publications (e.g. Farrell et al. (2020)), only the basic terminology is introduced here as well as the wake shape. In
the present work of this paper, the model has been extended with the calculation of added turbulence as proposed by Crespo and Hernández (1996). The power calculation has been extended by the $\cos(\gamma)^{p_p}$ adaptation to the yaw angle (Medici, 2005) and an efficiency term $\eta$ for tuning (Gebraad et al., 2014). Figure 1 depicts an illustration of the wake with its three areas: the potential core, the near wake area and the far wake area. For all areas a reduction factor $r = \Delta u / u_\text{free}$ can be calculated, where $u_\text{free}$ is the free wind speed and $\Delta u$ is the wind speed deficit. The potential core is a region from jets in a coflow (Lee
and Chu, 2003). Here, it is used to approximate the immediate region behind the rotor plane. Within the potential core, $r$ is constant. In the near and far field area $r$ reduces to 0, following a Gaussian shape with the extremum at the centerline or border of the potential core. The recovery rate is based on $\sigma_y$ and $\sigma_z$ in the respective crosswind directions. The potential core width

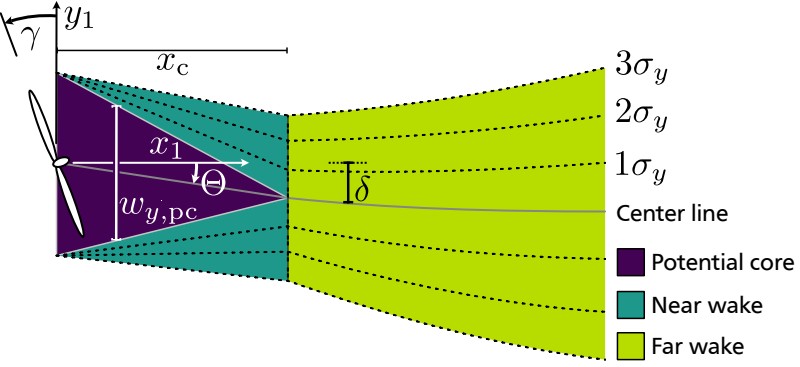

**Figure 1.** Sketched shape of the wake with the different sections, the deflection and areas of equal relative reduction by the Gaussian shape.

is described by $w_{y,\mathrm{pc}}$ and $w_{z,\mathrm{pc}}$, which continuously decrease for the length of the potential core $x_c$. Lastly, the deflection $\delta$ returns the position of the centerline.

The mentioned variables are dependent on turbine states, such as the thrust coefficient $C_\mathrm{T}$ and the yaw angle $\gamma$, the ambient turbulence intensity $I_0$ and a set of ten parameters. The parameters adjust wake properties such as the recovery rate, the expansion rate, the sensitivity to added turbulence levels and the influence of the yaw angle. The values of the parameters are listed in Table 1 in Section 3.

## 2.2 The Zone FLORIDyn model

An initial FLORIDyn model was published in Gebraad and van Wingerden (2014). The model is based on the previously published Zone FLORIS model which approximates the wake shape with three zones: near field, far field and mixing zone (Gebraad et al., 2014). Every zone has a formulation of the velocity recovery in downstream direction. To introduce dynamics, Observation Points (OPs) are created at the rotor plane at each time step. The OPs serve the purpose to describe the local FLORIS wake characteristics at their location. To do that, they inherit the turbine states at the time of their creation which are necessary to calculate the FLORIS wake. With time, each OP travels downstream, representing a mass of air traveling in the wind. Their travel path is determined by the borders of the FLORIS wake zones. The speed they travel with is equal to the effective wind speed they represent. Figure 2 shows the basic concept. Initial OPs are colored black to stress that they inherited the same state. The OPs created after the yaw step are colored white, showing that their inherited state differs.

With this framework, the steady-state wake represents the known FLORIS wake, but other than in FLORIS, changes propagate through the wake instead of instantly affecting turbines downstream. If, for instance, the yaw angle of the turbine changes, the new generation of OPs will inherit the new angle while old OPs still travel according to the previous angle.

In the case of overlapping wakes, an OP travels into the wake of another turbine. The OP locates the closest up- and downstream OPs from the foreign wake and interpolates their reduction factor at its location. In this model, the resulting

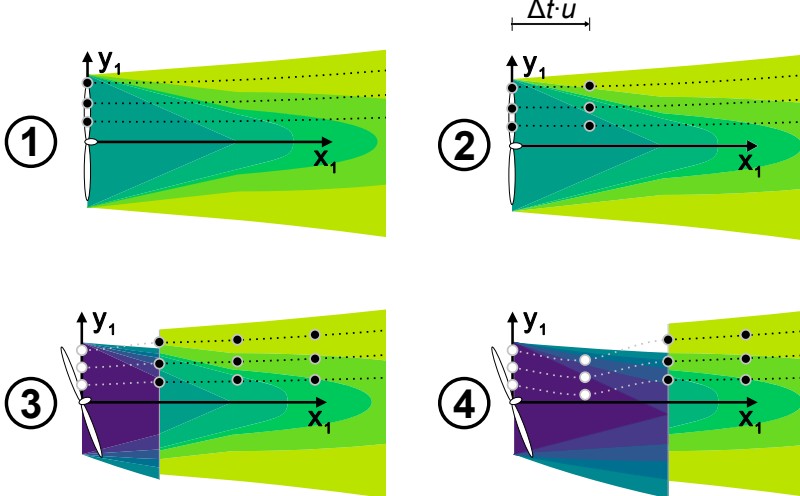

**Figure 2.** Creation and propagation of the OPs: In (1) a set of OPs is created, inherits the turbine state and travels downstream, following the FLORIS wake shape, shown in (2). In (3) the turbine state changed and the new OPs inherit a different state (now colored white) and follow the new, dark indicated, wake shape (4).

reduction of the free wind speed is calculated as follows:

$$u_{\text{eff,OP}}(u_{\text{free,OP}}, r_{\text{own}}, r_{\text{f,OP}}) = u_{\text{free,OP}}(1 - r_{\text{own}})\underbrace{\prod_{i=1}^{n_{\text{T}}}(1 - r_i)}_{r_{\text{f,OP}}} \tag{1}$$

where $u_{\text{free,OP}}$ is the free wind speed at the OP's location. This wake interaction model could also be exchanged for another formulation. The wind speed reduction $r_{\text{own}}$ is based on the OP's own wake and $r_i$ is the interpolated reduction of one of the $n_{\text{T}}$ upwind turbines.

To calculate the effective wind speed at the rotor plane the model calculates an effective velocity reduction factor $r_{\text{T}}$ for every turbine at every time step. The algorithm combines the reduction of each upstream turbine by a root-sum-square. Within one wake, the reduction factors of the zones are summed, weighted by the overlapping area with the rotor plane.

## 2.3 Changes to the FLORIDyn approach

Due to the changed underlying FLORIS model, the FLORIDyn approach needs to be adapted. Specifically, the move to a three-dimensional flow field requires a fitting distribution of the OPs, which is discussed in Section 2.3.1. This opens up the possibility to reformulate the calculation of the effective wind speed at the rotor plane, which is presented in Section 2.3.2. The travel speed of the OPs is addressed in Section 2.3.3. In this Section 2.3 we use the wake coordinate system $\mathcal{K}_1$, indicated by the lower index 1 (eg. $y_{1,\text{OP}}$). The relation between world and wake coordinate system will be explained in Section 2.4.

### 2.3.1 Distribution of the Observation Points

By changing the underlying FLORIS model the travel path of the OPs and their distribution has to be rethought. The Gaussian FLORIS model does not have defined borders and it is three-dimensional. To cover the crosswind wake area regularly for any number of OPs, an algorithm based on the sunflower distribution was used (Vogel, 1979). The algorithm returns a relative crosswind coordinate $(\nu_y, \nu_z) \in [-0.5, 0.5]$ for a given number of OPs. We used 50 OPs per time step. To cover the majority of the Gaussian wake influence, the wake width was chosen to be $\pm 3\sigma_y$ and $\pm 3\sigma_z$ from the centerline and the potential core. The following equation is used to calculate the position of an OP in the wake coordinate system:

$$y_{1,\mathrm{OP}}(\nu_{y,\mathrm{OP}}, \sigma_y, w_{y,\mathrm{pc}}, \delta) = \nu_{y,\mathrm{OP}}(6\sigma_y + w_{y,\mathrm{pc}}) + \delta \;, \tag{2}$$

$$z_{1,\mathrm{OP}}(\nu_{z,\mathrm{OP}}, \sigma_z, w_{z,\mathrm{pc}}) = \nu_{z,\mathrm{OP}}(6\sigma_z + w_{z,\mathrm{pc}}) \tag{3}$$

Note that this model only assumes a horizontal deflection. To add a vertical deflection, due to rotor tilt for instance, Equation (3) needs to be adapted accordingly. For simplicity's sake $\sigma_y$ is used, which represents $\sigma_{y,\mathrm{nw}}$ for $0 < x_1 \leq x_\mathrm{c}$ and $\sigma_{y,\mathrm{fw}}$ for $x_1 > x_\mathrm{c}$. Respectively, $\sigma_z$ is defined the same way. The variable $\delta$ describes the deflection of the centerline. If OPs travel below $z_1 = 0$ they are ignored. Since $\nu_y$ and $\nu_z$ are not changed during the simulation, they can be calculated a priori. They are then used in every time step for the new generation of OPs. OPs with the same relative coordinate follow each other and form what is called a chain. The number of chains is equal to the number of OPs created at each time step.

### 2.3.2 Wind speed at the rotor plane

Since OPs are created at the rotor plane and they interact with foreign wakes, they can be used to estimate the effective wind speed for the power generation. To do that, they have to be distributed across the rotor plane rather than the wake area:

$$y_{1,\mathrm{OP}}(\nu_{y,\mathrm{OP}}, \gamma) \,|_{x_1=0} = \nu_{y,\mathrm{OP}} D \cos\gamma \;, \tag{4}$$

$$z_{1,\mathrm{OP}}(\nu_{y,\mathrm{OP}}) \,|_{x_1=0} = \nu_{z,\mathrm{OP}} D \tag{5}$$

The next step is to determine the area represented by every OP. This is done offline by generating a Voronoi pattern (Voronoi (1908a), Voronoi (1908b)) with the OPs' relative location as seeds and a circular boundary with radius $0.5$. The area of the resulting polygons is normalized by the rotor area and used as weight. All weights are stored in the vector $\boldsymbol{w}$.

During the simulation, the OPs calculate the reduction of foreign wakes $r_{\mathrm{f,OP}}$ on themselves as shown in Equation (1). Stored in a vector $\boldsymbol{r}_\mathrm{f} = [r_{\mathrm{f},1}, \cdots, r_{\mathrm{f,nOP}}]^\top$ the effective wind speed at the rotor plane is calculated as follows:

$$u_{\mathrm{eff}} = \boldsymbol{w}^\top (\boldsymbol{r}_\mathrm{f} \circ \boldsymbol{u}) \tag{6}$$

where $\circ$ stands for the element-wise multiplication and $\boldsymbol{u}$ represents a vector of the free wind speeds at the locations of the OPs. An OP considers itself influenced by a foreign wake if the closest foreign OP is less than $\frac{1}{4}D$ away. This is an arbitrary chosen threshold to reduce the number of OPs for the interaction interpolation. As the outer wake OPs represent the most recovered sections of the wake, this still results in a smooth influence transition.

### 2.3.3 Travel speed

In the former version of the FLORIDyn model, the OPs travel with the effective wind speed they represent. Regions in the center of the wake with lower effective wind speeds therefore propagate the changes slower than the outer areas. While this seems an intuitive choice, it leads to problems. Initial simulation results showed that, in comparison to the SOWFA simulation, the effects of a state change arrive noticeably slower in FLORIDyn at downstream turbines. Also, due to the difference in OP travel speed, the outer regions adapt their shape earlier in a downstream location, which leads to overlapping areas with the slow regions, which have not adapted yet. This makes the wake representation not injective anymore: Multiple OPs occupy and describe the same space at the same time with varying properties.

In this article, the OPs are assumed to propagate with the speed of the freestream wind rather than the effective wind speed in accordance with Taylor's Frozen Turbulence Hypothesis (Taylor, 1938). The decision is supported by experimental results from Schlipf et al. (2010) and has also been used by other similar codes, e.g. Grunnet et al. (2010). This also solves the issue of the overlapping wake areas since neighboring OPs travel at the same speed and follow the same state changes. Another implication of this adaptation is that OPs no longer need to calculate the influence of foreign wakes at every time step. This would be used to determine their effective wind speed and thus how far they travel downstream in one time step. The only OPs which need to calculate the foreign influence are the ones at the rotor plane in order to determine the effective wind speed according to Equation (6). These model assumptions also significantly decrease the computational load during the simulation. The downside of the change is that the effects of state changes now arrive too fast and abrupt at downstream turbines, which will be seen and discussed with the simulation results in Section 3. In future work, the wake propagation speed could be a tuning parameter which is set depending on atmospheric conditions such as the turbulence intensity for instance (Andersen et al., 2017).

### 2.4 Including directional dependency and Observation Point propagation

In this section we address how the OPs, and therefore the wakes, react to a wind direction change. We assume that a wind direction change only affects the wake orientation and that the wake structure and downstream evolution (as defined by the underlying FLORIS model) can be seen independent from the free stream behavior. It is therefore possible to split the two aspects into two coordinate systems: The world coordinate system $\mathcal{K}_0$ and the wake coordinate system $\mathcal{K}_1$. The free flow conditions are described in $\mathcal{K}_0$, whereas the wake properties are described in $\mathcal{K}_1$. An OP links these two coordinate systems.

The underlying FLORIS model is described in $\mathcal{K}_1$, where the origin $x_1 = y_1 = z_1 = 0$ is located in the center of the rotor plane. The downwind distance is denoted as $x_1$, $y_1$ describes the horizontal crosswind distance and $z_1$ the vertical one. $\mathcal{K}_0$ does not have a special orientation apart from $z_0 = 0$ being the ground level and the $z_0$ axis pointing upwards. In this work, $x_0$ describes the West-East axis, $y_0$ the South-North axis. To transform a location vector $\boldsymbol{r}_1$, described in $\mathcal{K}_1$ of a turbine with the

rotor-center location $\boldsymbol{t}_0$, into $\boldsymbol{r}_0$ the rotational matrix $\mathbf{R}_{01}$ is used:

$$\boldsymbol{r}_0 = \begin{bmatrix} x_0 \\ y_0 \\ z_0 \end{bmatrix} = \boldsymbol{t}_0 + \mathbf{R}_{01}(\varphi)\boldsymbol{r}_1 = \begin{bmatrix} x_{0,\mathrm{T}} \\ y_{0,\mathrm{T}} \\ z_{0,\mathrm{T}} \end{bmatrix} + \begin{bmatrix} \cos\varphi & -\sin\varphi & 0 \\ \sin\varphi & \cos\varphi & 0 \\ 0 & 0 & 1 \end{bmatrix} \begin{bmatrix} x_1 \\ y_1 \\ z_1 \end{bmatrix} \tag{7}$$

This equation assumes a uniform wind direction $\varphi$ at every location. This will not be the case for the formulation used for the OP propagation later on in Eq. (9). Each OP has two location vectors, $\boldsymbol{r}_{0,\mathrm{OP}}$ and $\boldsymbol{r}_{1,\mathrm{OP}}$, one for each coordinate system. The OP's position update and its reduction factor is calculated in $\mathcal{K}_1$. $\mathcal{K}_0$ is used to calculate the wake interaction and to determine the wind speed, the wind direction and the ambient turbulence intensity. At the OP's creation, $\boldsymbol{r}_{1,\mathrm{OP}}$ is determined by the Equations (4) and (5) for the crosswind coordinates, the downwind coordinate is set to 0. Its world location, $\boldsymbol{r}_{0,\mathrm{OP}}$, is
then determined by Equation (7) with the wind direction $\varphi_{0,\mathrm{T}}$ at the turbines location. To iterate the location of an OP from time step $k$ to time step $k+1$ the downwind step is calculated first in $\mathcal{K}_1$:

$$x_{1,\mathrm{OP}}(k+1) = x_{1,\mathrm{OP}}(k) + u_{\mathrm{OP}}\,\Delta t \tag{8}$$

where $\Delta t$ is the time step duration and $u_{\mathrm{OP}}$ is the magnitude of the wind vector $\boldsymbol{u}_{0,\mathrm{OP}}$ at the OPs location $\boldsymbol{r}_{0,\mathrm{OP}}$. The direction will be applied in Eq. (9). For the scope of this work, $\boldsymbol{u}_{0,\mathrm{OP}}$ can only have non-zero components in $x_0$ and $y_0$ direction. With
$x_{1,\mathrm{OP}}(k+1)$ the new crosswind locations $y_{1,\mathrm{OP}}(k+1)$ and $z_{1,\mathrm{OP}}(k+1)$ can be calculated with the Equations (2) and (3), respectively. This completes the transition $\boldsymbol{r}_{1,\mathrm{OP}}(k) \to \boldsymbol{r}_{1,\mathrm{OP}}(k+1)$. Note that only $x_{1,\mathrm{OP}}(k)$ is needed to determine the OP's location in $\mathcal{K}_1$. At the cost of calculating $y_{1,\mathrm{OP}}(k)$ and $z_{1,\mathrm{OP}}(k)$ again at each time step, they do not have to be stored as states. To update $\boldsymbol{r}_{0,\mathrm{OP}}(k)$ the step which the OP took in $\mathcal{K}_1$ has to be translated into $\mathcal{K}_0$:

$$\boldsymbol{r}_{0,\mathrm{OP}}(k+1) = \boldsymbol{r}_{0,\mathrm{OP}}(k) + \mathbf{R}_{01}(\varphi_{0,\mathrm{OP}})[\boldsymbol{r}_{1,\mathrm{OP}}(k+1) - \boldsymbol{r}_{1,\mathrm{OP}}(k)] \tag{9}$$

where $\varphi_{0,\mathrm{OP}}$ is the wind direction at $\boldsymbol{r}_{0,\mathrm{OP}}(k)$. Note that $\varphi_{0,\mathrm{OP}}$ refers to one OP's individual wind direction, other OPs may have different values. This means that each OP propagates on its own and non-uniform wind directions can be simulated. Figure 3 shows the OP step in the wake and world coordinate system. In Subfigure 3-1 and 3-2 the wind direction is constant, indicated by the arrow left to the $y_0$ axis. The OP calculates its step in the wake coordinate system (dotted arrow) and updates its location vectors. These are here simplified to $\boldsymbol{r}_0$ and $\boldsymbol{r}_1$. In 3-3 the wind direction changes and the former FLORIS wake description
is invalid and greyed out. With the new wind direction $\mathbf{R}_{01}(\varphi_{0,\mathrm{OP}})$ is calculated differently. The OP can calculate its step in the wake coordinate system as before, but its translation $\mathcal{K}_1 \to \mathcal{K}_0$ changed. Note that neither $\boldsymbol{r}_0$ nor $\boldsymbol{r}_1$ are influenced by the changed wind direction. Their magnitude and orientation remain the same in their respective coordinate systems, however, their orientation towards each other changes.

### 2.5   Calculation of $C_\mathrm{T}$ and $C_\mathrm{P}$

The thrust coefficient $C_\mathrm{T}$ is often approximated following the actuator disc theory: $C_\mathrm{T}(a) = 4a(1-a)$, where $a$ is the axial induction factor. To circumvent this approximation, simulations or experiments can be used to create look-up tables. Since

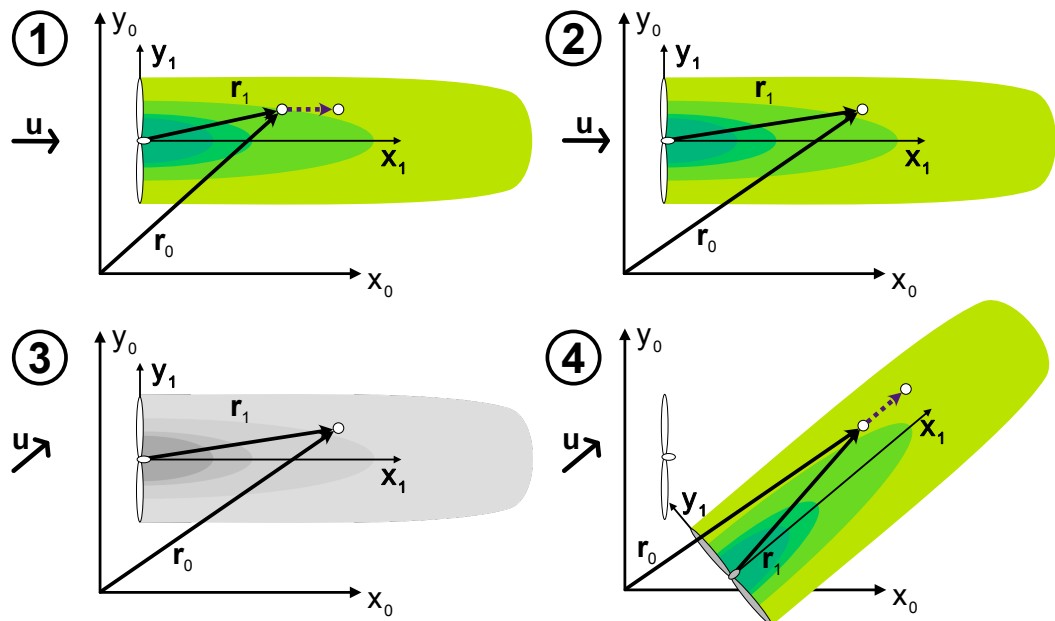

**Figure 3.** This figure visualizes the working of Equation (9), which is applied for each OP individually. In $(1 \rightarrow 2)$, the position update of an OP in a time step with a constant wind direction is depicted. $(3 \rightarrow 4)$ shows the position update when the wind direction changes. In this case, the wake coordinate system is rotated around the OP's location to match the new downstream direction. This causes the apparent origin of the wake in the world coordinate system to change, which is visualized by the gray turbine.

most equations of the Gaussian FLORIS model are dependent directly on $C_T$ rather than $a$, we used look-up tables generated in SOWFA to align FLORIDyn's thrust coefficient with what the turbines in the validation environment experience. For completeness, we also use look-up tables for the power coefficient $C_P$. The tables are generated for the DTU10 MW reference

turbine (Bak et al., 2013). It has to be added that these tables are generated from a grid of high fidelity simulations, where the coefficients were read after the simulation converged to a steady-state. The tables can therefore only approximate the effect a changing turbine state and changing wind field conditions onto $C_T$ and $C_P$. Control approaches for axial-induction-based controllers, such as the one presented by Annoni et al. (2016), successfully use similar look-up tables, which is why we assume these to be sufficient. Nevertheless, an extension for dynamic circumstances would be a valuable addition for future work, but

is also connected to a significant computational effort.

In the tables, the coefficients are described dependent on the blade pitch angle $\beta$ and the tip-speed-ratio $\lambda(\omega, u_{\text{eff}})$, where $\omega$ is the angular velocity of the rotor. However, neither FLORIS nor FLORIDyn can provide $\lambda$ and $\beta$. What they can provide is $u_{\text{eff}}$. Combined with the assumption that each turbine follows a greedy control strategy and maximizes $C_P(\lambda, \beta)$ for the given wind, we can formulate the coefficients dependent only on $u_{\text{eff}}$: first, maximize $C_P$ within the physical limitations of the wind

turbine for all wind speeds, then use the $\lambda_{P,\text{max}}$ and $\beta_{P,\text{max}}$ to calculate the respective $C_T$. The resulting curves can be seen in Figure 4.

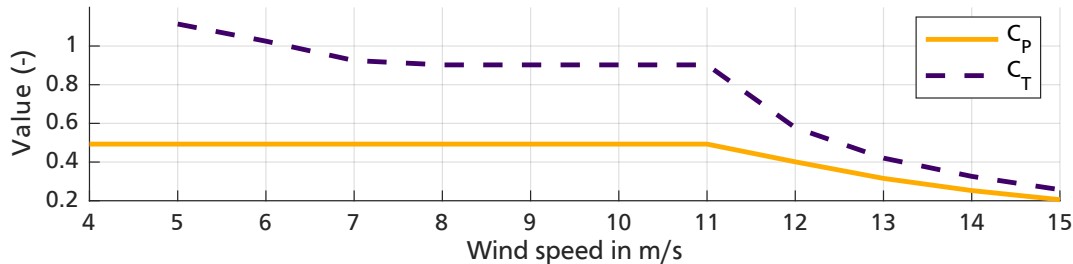

**Figure 4.** Greedy control settings of the un-yawed 10 MW DTU reference turbine based on the effective rotor wind speed.

Unfortunately, the resulting $C_T(u_{\text{eff}})$ values can get very high, especially for low wind speeds. This conflicts with some FLORIS equations which comprise the term $\sqrt{1 - C_T}$ and become complex for $C_T$ values above 1. To avoid these issues, $C_T(u_{\text{eff}})$ is limited to its value at the Betz limit: $C_T|_{a=1/3} = 0.\bar{8}$ (Bianchi et al., 2007). Another complication is the calculation of the added turbulence levels as it is the only equation which requires the axial induction factor. In this case, the calculation of $C_T(a)$ was inverted to determine $a(C_T)$, based on the actuator disc theory, as follows:

$$a = \frac{1}{2}\left(1 - \sqrt{1 - C_T}\right).$$
(10)

Yaw effects on $C_T$ and $a$ are neglected here. In future work this expression could be substituted, for instance by the polynomial approximation of Madsen et al. (2020). It extends $a(C_T)$ to $C_T$ values above 1. However, as $C_T$ is limited in this work, this extension is not necessary. The power coefficient is the remaining aspect which was used unaltered from the lookup tables. For the tested wind speeds below $11\,\text{ms}^{-1}$ the power coefficient is constant at $C_P = 0.4929$. The effect of $\gamma$ is approximated by multiplying $C_P$ with $\cos(\gamma)^{p_p}$. For simplicity's sake we assume $p_p$ to be a constant value. This could be extended by the work presented by Liew et al. (2019), which takes the presence of other wakes into account. Similarly, Howland et al. (2020), presents an adaptation for locally varying wind profiles.

## 2.6 Wind field model

In order to drive the FLORIDyn model, the wind field needs to be able to simulate heterogeneous, changing environmental conditions. The implemented solution is inspired by the work of Farrell et al. (2020). The basic assumption is that measurements of the wind field variables are available at certain locations. This could be due to satellite data, LiDAR measurements, met masts or other sensors. The value of a measurement for the location of an OP is then interpolated between the measurements available. To reduce the computational effort of an interpolation at every time step, a nearest neighbor interpolation (NNI) is desirable. To get a sufficient resolution of the measurements to justify a NNI the sparse measurements $\boldsymbol{m}$ have to be mapped to dense measurement grid points $\boldsymbol{m}_{\text{g}}$:

$$\boldsymbol{m}_{\text{g}} = \mathbf{M}\boldsymbol{m}$$
(11)

where the matrix $\mathbf{M}$ describes the mapping and can be calculated offline. The $i$-th row in $\mathbf{M}$ describes the percental composition of $\boldsymbol{m}_{\text{g},i}$ from $\boldsymbol{m}$. As a result, the sum of every row in $\mathbf{M}$ is equal to 1. This way, a more complex interpolation can be reduced

**Table 1.** Parameters used in the simulation with the values they influence

| FLORIS | | | | | | | | | | FLORIDyn | | Wind |
|---|---|---|---|---|---|---|---|---|---|---|---|---|
| Wake expansion | | Added turbulence | | | | Potential core | | Power | | Chains, OPs | | Shear |
| $k_a$ | $k_b$ | $k_{f,a}$ | $k_{f,b}$ | $k_{f,c}$ | $k_{f,d}$ | $\alpha^*$ | $\beta^*$ | $\eta$ | $p_p$ | $n_\mathrm{c}$ | $n_\mathrm{OP}$ | $\alpha_s$ |
| 0.38371 | 0.003678 | 0.73 | 0.8325 | 0.0325 | $-0.32$ | 2.32 | 0.154 | 0.8572 | 2.2 | 50 | 200 | 0.08 |

to a matrix multiplication and a NNI at runtime. In this work, a linear interpolation is used to map the measurements to the grid points, which are spaced in a $20 \times 20\,\mathrm{m}$ grid. OPs outside of the grid defined by $\boldsymbol{m}_\mathrm{g}$ use the closest grid point. This method is also independent from the quantity measured. In this work, the wind speed, the wind direction and the ambient turbulence intensity were interpolated with the presented method.

However, the presented method is only meant for values changing in $x_0$ and $y_0$ direction. The wind speed is the only field measurement which is also changed in the $z_0$ direction, wind direction and ambient turbulence intensity is assumed to be constant in vertical direction. Following Farrell et al. (2020) the power law is applied:

$$u(z_0) = \left( \frac{z_0}{z_{0,\mathrm{m}}} \right)^{\alpha_s} u(z_{0,\mathrm{m}}) \tag{12}$$

where $z_{0,\mathrm{m}}$ is the height of the measurement and $\alpha_s$ is the shear coefficient. The shear coefficient approximates the combined effect of atmospheric stability and surface roughness. A small value describes unstable flow conditions. Examples for characteristic $\alpha_s$ values due to surface roughness are: $0.11$ over water, $0.16$ over grass, $0.20$ over shrubs, $0.28$ over forests and $0.40$ over cities (Emeis, 2018). In this work $z_{0,\mathrm{m}}$ is equal to the hub height $z_\mathrm{h}$ of the turbine.

## 3 Simulation results

In this section, the Gaussian FLORIDyn model is compared to SOWFA with the focus on turbine interaction. Two wind farm layouts are considered for comparison: Three consecutive turbines and a nine turbine cluster arranged in a $3 \times 3$ configuration. The DTU 10 MW reference turbine is used for all simulations. Table 1 summarizes the FLORIS and FLORIDyn parameters used in the simulations. The FLORIS parameters $k_a$ and $k_b$ are from Niayifar and Porté-Agel (2015) , $k_{f,a}$ to $k_{f,d}$ are set based on FLORISSE_M (Github DCSC, 2020) and $\alpha^*$, $\beta^*$ follow the findings of Bastankhah and Porté-Agel (2016). The efficiency $\eta$ was tuned based on turbine T0 in the three turbine baseline case, $p_p$ was tuned based on the three turbine yaw case (Section 3.1.1 and 3.1.2 respectively). For FLORIDyn, $n_\mathrm{c}$ relates to the number of OP chains per turbine and $n_\mathrm{OP}$ to the number of OPs per chain. The value of $n_\mathrm{OP}$ was set to cover the entire relevant downstream domain of a turbine, $n_\mathrm{c}$ was set to maintain a sufficient density of OPs at the location of other turbines. In FLORIDyn, one time step is $4.0$ s long. Table 1 also includes the wind shear coefficient, $\alpha_s$, which was approximated based on the free flow in SOWFA. The inflow boundary conditions for SOWFA are provided by a precursor simulation which simulates a horizontally homogenous, conventionally neutral atmospheric boundary layer including Coriolis effects. The SOWFA settings differ for the three turbine case and the nine turbine case and will be explained in the respective sections.

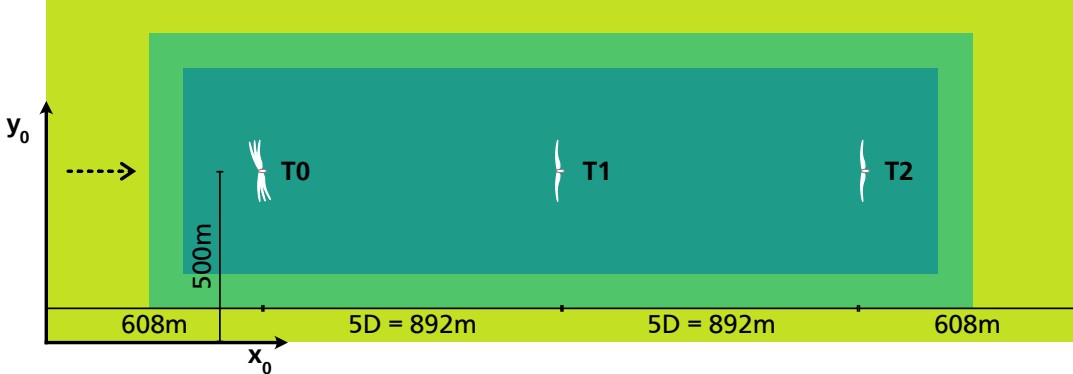

**Figure 5.** Scaled layout of the three turbine case with the wind direction indicated by an arrow on the left. The $0°$, $10°$ and $20°$ yaw orientations from T0 are indicated as turbine symbols with the according orientation. The colored background areas indicate the zones of cell refinement.

## 3.1 Three turbine case

The three turbines are placed $5D = 892$ m apart from each other in downwind direction. Turbine T0 is located at $(608 \text{ m}, 500 \text{ m})$ and T1 and T2 are at $(1500 \text{ m}, 500 \text{ m})$ and $(2392 \text{ m}, 500 \text{ m})$ respectively. The mean wind speed at hub height is approximately 315  $8.2 \text{ ms}^{-1}$ with an ambient turbulence intensity of roughly $6 \%$. The mean wind direction is constant along the x axis. The full SOWFA flow field domain spans $3 \times 1 \times 1$ km and was simulated with a time step of $\Delta t = 0.04$s. The base cells of the flow field are $10 \times 10 \times 10$ m. The refinement areas are centered in the domain and have no offset from the ground. The first refinement is $2.4 \times 0.8 \times 0.5$ km with $5 \times 5 \times 5$ m cells, the second one is $2.2 \times 0.6 \times 0.35$ km with $2.5 \times 2.5 \times 2.5$ m cells. Figure 5 shows the to-scale layout including the areas of cell refinement. In SOWFA, the turbines are modelled with the built-in Actuator Line 320  Method (ALM) implementation (Sorensen and Shen, 2002).

To give a better idea of the low frequency, less-turbulent dynamics, the power generated in SOWFA is also presented filtered by a zero-phase second-order low-pass filter. This non-causal filter is added to aid the visual interpretation of the simulation results. The filter has a damping ratio of $d = 0.7$ and a natural frequency of $\omega = 0.03 \text{ s}^{-1}$. This allows for a more equal comparison as FLORIDyn is sampled at a lower frequency and turbulences are only included as a flow field metric.

A regular second-order low-pass filter with the same $d$ and $\omega$ is used for the FLORIDyn data. This causal filter visualizes how low-pass filtering would effect the predicted power generated. This could have advantages due to the changes made to the OP travel speed in Section 2.3.3 which can lead to a very abrupt wake interaction, as will be discussed in Section 3.1.1. However, the filter also naturally adds a phase shift to the data, an effect which might not be desired.

Note that the two filters have different purposes: The non-causal SOWFA filter aims to help to interpret the simulation 330  results, while the causal FLORIDyn filter explores if and when the use of a low-pass filter would be advantageous or if it would decrease the quality of the results.

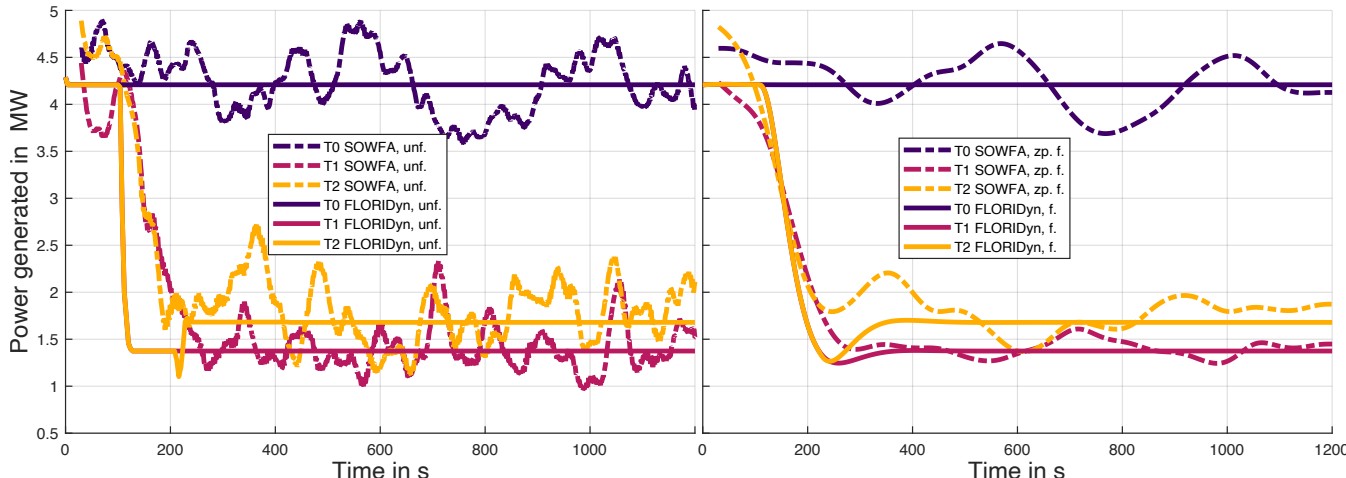

**Figure 6.** Wind farm start-up and steady-state comparison of the power generated in SOWFA and in FLORIDyn. The unfiltered data is plotted in the left figure, filtered data on the right. The SOWFA data is filtered with a zero-phase (noncausal) low-pass filter, FLORIDyn with a causal low-pass filter.

### 3.1.1 Comparison of the wind farm start-up and steady-state

In Figure 6 the power generated by the turbines in FLORIDyn is compared to the SOWFA simulation. The dynamics in this simulation are the turbulent wind field and the settling of the wake. In the unfiltered data, the interaction in FLORIDyn sets

in earlier and more abrupt than in SOWFA. This is due to the OPs traveling at the free wind speed, as explained in Section 2.3.3. The slight curvature of the drop at $t \approx 100\,\mathrm{s}$ can be explained by the wind shear: OPs at a lower altitude travel at a slower free wind speed than OPs at a higher altitude and therefore arrive later at the downstream turbine and therefore affect the turbine later. There are two major aspects to address in order to close the gap between the SOWFA and the FLORIDyn start-up: on the one hand, the way state changes propagate through the wake; on the other hand, how a downstream turbine

reacts to the new wind field. To give an idea how a change of the latter aspect would influence the plot, Figure 6 also shows low-pass filtered FLORIDyn data in comparison to zero-phase filtered SOWFA data. The FLORIDyn data aligns much more with the SOWFA data but still shows discrepancies in terms of dynamic response and steady-state quality of the solution. It should be emphasized that the in FLORIDyn applied filter does not affect the wake, it only adds an artificial dynamic response to the power calculation. This is important when heterogeneous and changing wind directions are taken into account.

The power generated after the wind farm start-up remains steady in FLORIDyn. This is since there are no turbulent wind speed changes in FLORIDyn. In this state, FLORIDyn is equal to the underlying FLORIS model, therefore errors in this state need to be solved by adapting the FLORIS model. This could be done by parameter tuning, which has only been done partially in this work ($\eta$ and $p_p$ see introduction Sec.3).

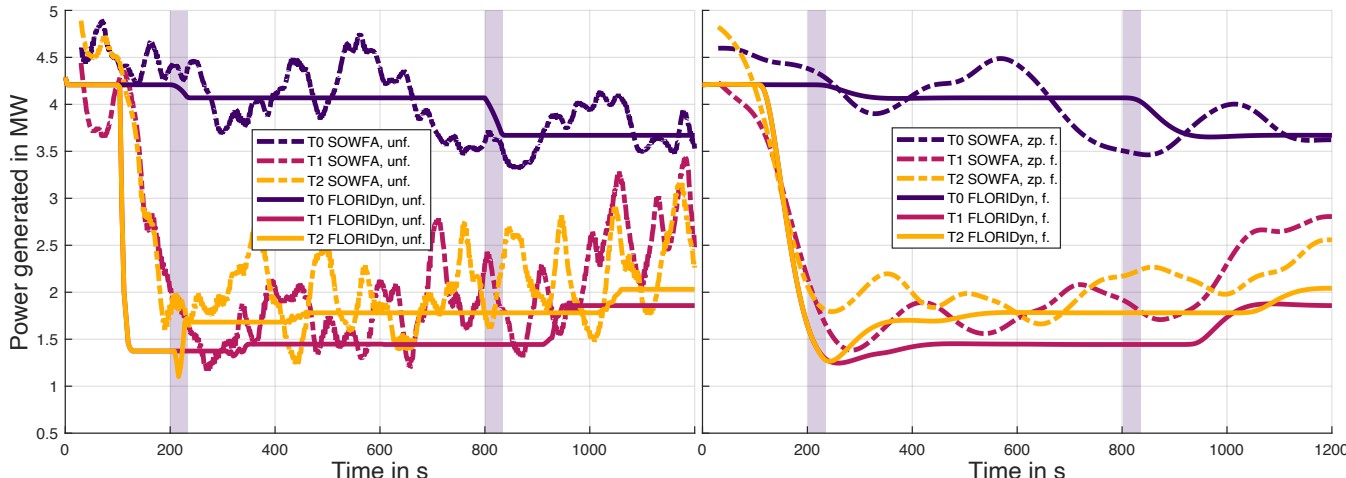

**Figure 7.** Comparison of the power generated in SOWFA and in FLORIDyn with changing yaw angles. The transparent bars indicate the time window in which turbine T0 increases its yaw angle by $10°$. The left plot shows the unfiltered data, the right one the filtered data. The SOWFA data is filtered with a zero-phase (noncausal) low-pass filter, FLORIDyn with a causal low-pass filter.

To incorporate the turbulent wind speed changes, at least to a certain degree, an estimation of the wind speed at the turbine location would be necessary. This could be done by including wind speed sensor data or estimating the wind based on the power generated (Gebraad et al., 2015) or based on a torque balance equation (Ortega et al., 2013).

A notable aspect of this simulation is the influence of the added turbulence. Because T0 adds turbulence to the wind field, the wake of turbine T1 recovers faster. Turbine T2 thus experiences higher wind speeds and generates more power than it would without the additional turbulence. This effect can also be observed in the SOWFA data. The old Zone FLORIDyn model is not able to capture this effect due to the underlying Zone FLORIS model. It shows how the FLORIDyn framework is inherently dependent on the capabilities of the employed FLORIS model.

### 3.1.2 Comparison during a yaw angle change

In this simulation, the yaw angle $\gamma$ of turbine T0 is changing from $0°$ to $20°$ in steps of $10°$, starting at $t = 200\,\text{s}$ and $800\,\text{s}$. The change rate of $\gamma$ is set to $0.3°\,\text{s}^{-1}$. Figure 7 shows the unfiltered SOWFA data in comparison to the unfiltered FLORIDyn data on the left, as well as the filtered data on the right. Filtering was performed as described in the introduction of Section 3.1.

In FLORIDyn, turbine T1 shows a slight reaction to the yaw changes of turbine T0 at roughly $t \approx 320\,\text{s}$ and more significantly at $t \approx 920\,\text{s}$. The influence of the state change then travels further and impacts T2 at $t \approx 430\,\text{s}$ and, as well more significantly, at $t \approx 1030\,\text{s}$. In SOWFA, the reaction is obscured by turbulent influences. However, an increase in average power can be seen for T1 and T2 throughout the entire simulation. Figure 8 shows the baseline simulation in comparison to the SOWFA simulation, in absolute values and the difference. The data of both simulations can be compared since they use the same wind field. It allows for a more accurate determination of the reaction time to the upstream change. Turbine T1 starts

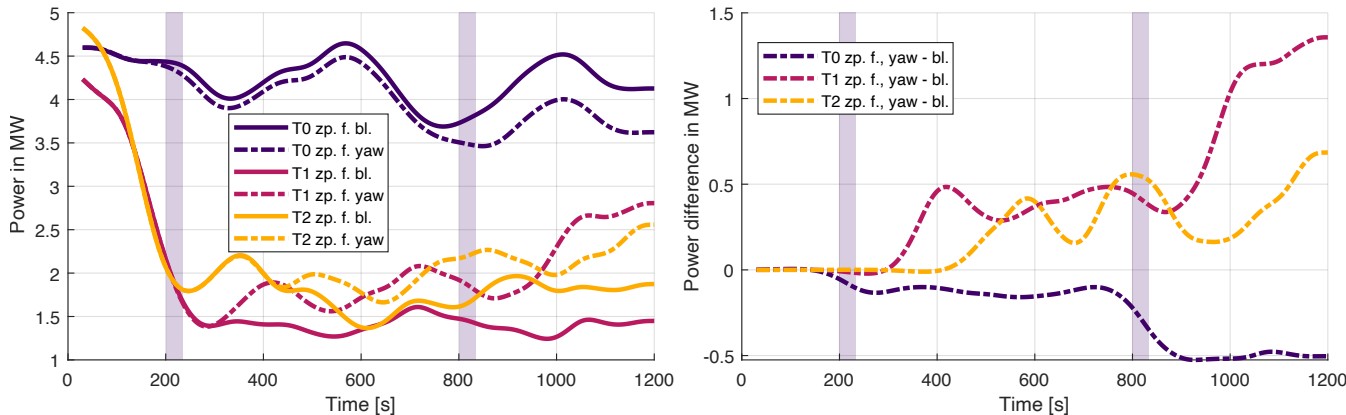

**Figure 8.** Comparison between the zero phase filtered (zp.f.) SOWFA data in the baseline case (bl.) and in the yaw case. On the left in absolute values, on the right with the difference between the yaw case and the baseline case. The transparent bars indicate the time window in which turbine T0 increases its yaw angle, first from $0°$ to $10°$, then to $20°$.

to react to the first yaw angle change at $t \approx 320 \pm 8$ s, T2 at $t \approx 434 \pm 8$ s. Given the distance to T0, this translates to a travel speed of the first influence between $[6.98, 7.97]\,\mathrm{ms}^{-1}$ to T1 and $[7.38, 7.90]\,\mathrm{ms}^{-1}$ to T2. This indicates that first effects of the yaw angle change do travel at almost free stream velocity and the times align with the FLORIDyn prediction. However,

FLORIDyn does lack the dynamic nature of the interaction, that means the response of the wake to the state change of the upstream turbine and the response of the downstream turbine to changes in the wake. Given that all OPs travel at their free stream velocity, turbine state changes are directly picked up by the OPs, transported and the FLORIDyn turbine reacts immediately when the OP arrives. The low-pass filtered results provide an idea how a dynamic response could change the results. The unfiltered difference between the SOWFA simulations is given in the Appendix A1. Figure 7 also shows that FLORIDyn

underestimates the gain in generated energy in the steady-state region. The error likely lies with the underlying FLORIS model as it is a steady-state error. Better parameter tuning would likely decrease or even eliminate the error.

### 3.2 Nine turbine case

In order to test the model in a changing environment, a simulation with nine turbines was performed. The turbines are arranged in a three by three grid with $900\,\mathrm{m}$ distance to the closest turbines and $600\,\mathrm{m}$ to the edge. The setup is presented in Figure

9 as well as the numbering of the turbines. The wind field performs a $60°$ uniform wind direction change from $15°$ to $75°$, as indicated in Figure 9. The change starts at $t = 600$ s with $0.2°\,\mathrm{s}^{-1}$ and ends at $t = 900$ s. The change in wind direction is achieved by using SOWFA's built-in utility to specify the wind speed and wind direction at a certain height and time. For the remainder of the simulation, the wind field conditions remain steady. To keep the computational load of the SOWFA simulation low, the DTU 10 MW turbines were simulated with the Actuator Disc Method (ADM). This also allows a coarser grid and

time resolution: The domain is discretized in $10 \times 10 \times 10$ m cells and the SOWFA time step length is set to $0.5$ s. The flow field spans $3 \times 3 \times 1$ km. The average wind speed during the simulation is $8.2\,\mathrm{ms}^{-1}$ and the ambient turbulence intensity is

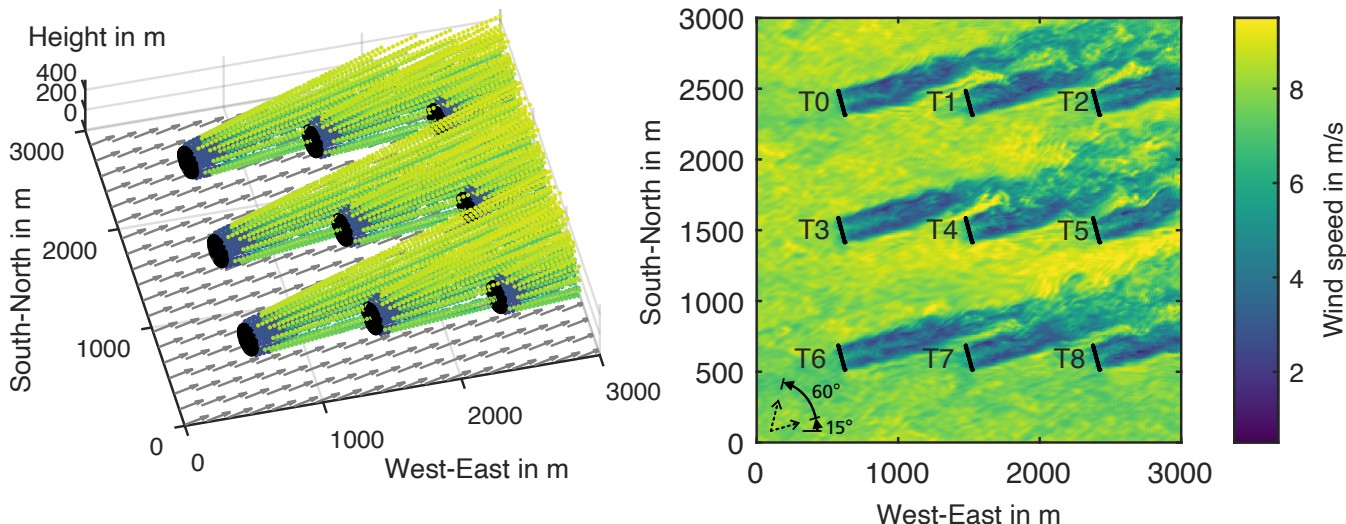

**Figure 9.** Complete nine turbine FLORIDyn flow field in comparison to SOWFA at $t = 600$ s. The wind direction change is indicate in the lower left corner of the SOWFA plot.

approximately 6 %. During the simulation, the turbines maintain a yaw angle of $0°$ and turn with the wind. For simplicity we assume ideal wind direction tracking capabilities and apply a prescribed motion. For more information see the dataset which contains the SOWFA files for the case and the precursor simulation (Becker, 2022b).

Figure 10 shows the flow field during the wind direction change, starting at the time instances $t = 700$ s, $t = 800$ s and 900s. The SOWFA slices are taken at hub height. To show the center of the FLORIDyn flow field, only OPs between $0.5\,z_\mathrm{h} < z_\mathrm{OP} < 1.5\,z_\mathrm{h}$ are plotted. As the wake expands, more chains leave these bounds which leads to a sparser description of the wake in the plot. Due to the changes to FLORIDyn described in Section 2.3.3, the OPs do not influence each other in the field and OPs with a higher velocity can appear among OPs with a lower velocity. The net effect of multiple OPs is only calculated at the rotor
plane. The grey arrows in the FLORIDyn plots indicate the current wind direction. The plots of both simulations visualize how the wakes slowly transition to the new wind direction, forming a bow-shape in the process. FLORIDyn seems to describe the general path of the SOWFA wakes quite well. It also capture some effects like shorter, wider wakes of T4 and T5 at $t = 900$ s.

To more accurately judge the timing of FLORIDyn, Figure 11 shows the generated power of all nine turbines. The plots are arranged in the same way as the turbines in the flow field plots. All plots show the filtered and unfiltered data of FLORIDyn and
SOWFA. The filtering is identical to the filtering in Section 3.1. The grey area marks the time window of the wind direction change. Looking at the magnitude of the generated power, FLORIDyn predicts the average of the free stream turbines T0, T3, T6, T7 and T8 quite well, but the remaining turbines show a noticeable offset in generated power. This could be due to speed up effects and is briefly discussed in Appendix A2. The interesting aspect is the timing of the wake interaction from upstream turbines with downstream turbines. The generated power by T4 shows the passing of the wake of T6 during the
wind direction change. Noticeable is the accuracy with which the unfiltered FLORIDyn data aligns the unfiltered SOWFA

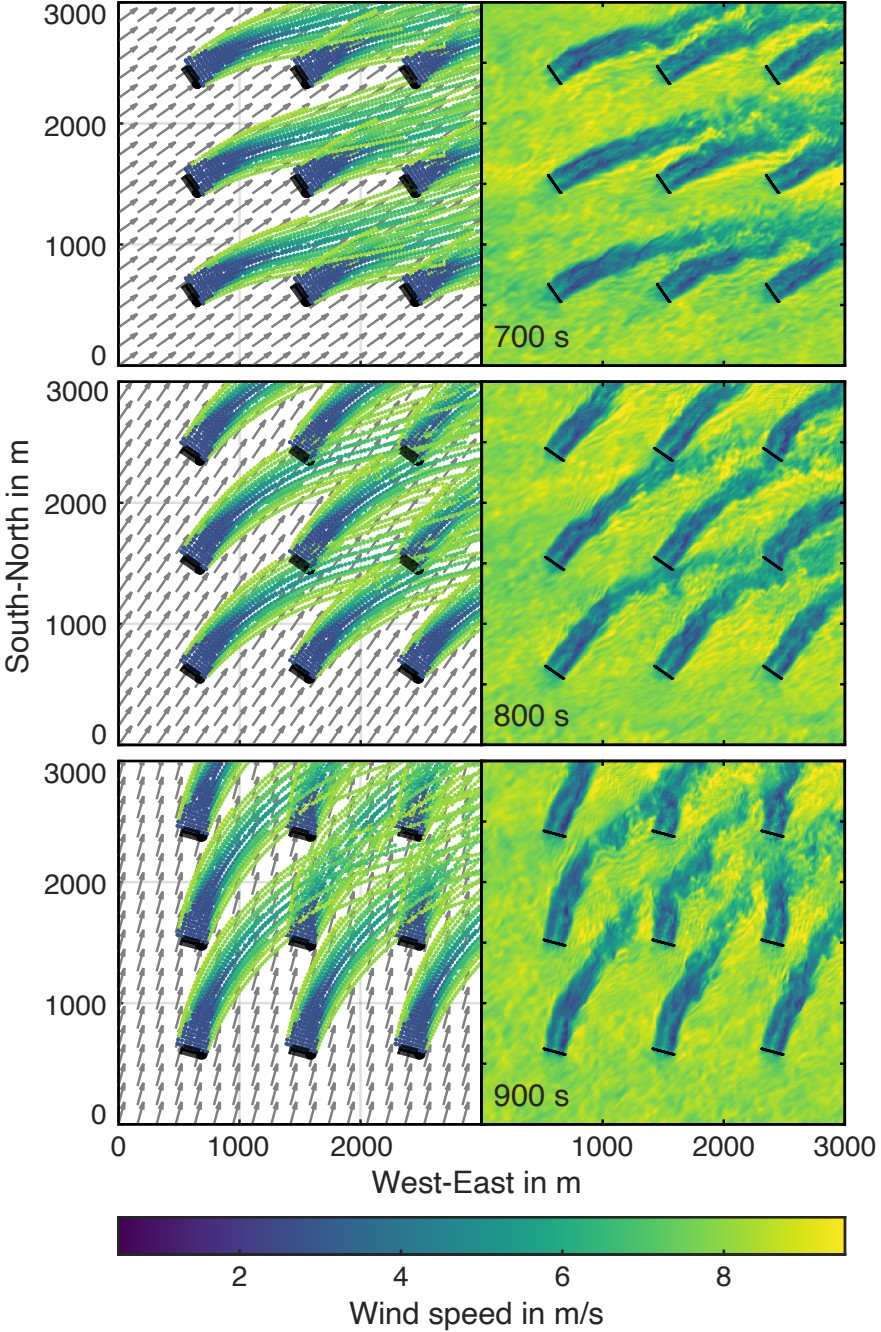

**Figure 10.** Nine turbine flow field at hub height during the wind direction change at $t = 700\,\text{s}$ (top), $t = 800\,\text{s}$ (center) and $t = 900\,\text{s}$ (bottom). The FLORIDyn flow field is on the left and includes grey arrows as an indicator of the current wind direction. On the right is the corresponding snapshot from the SOWFA simulation.

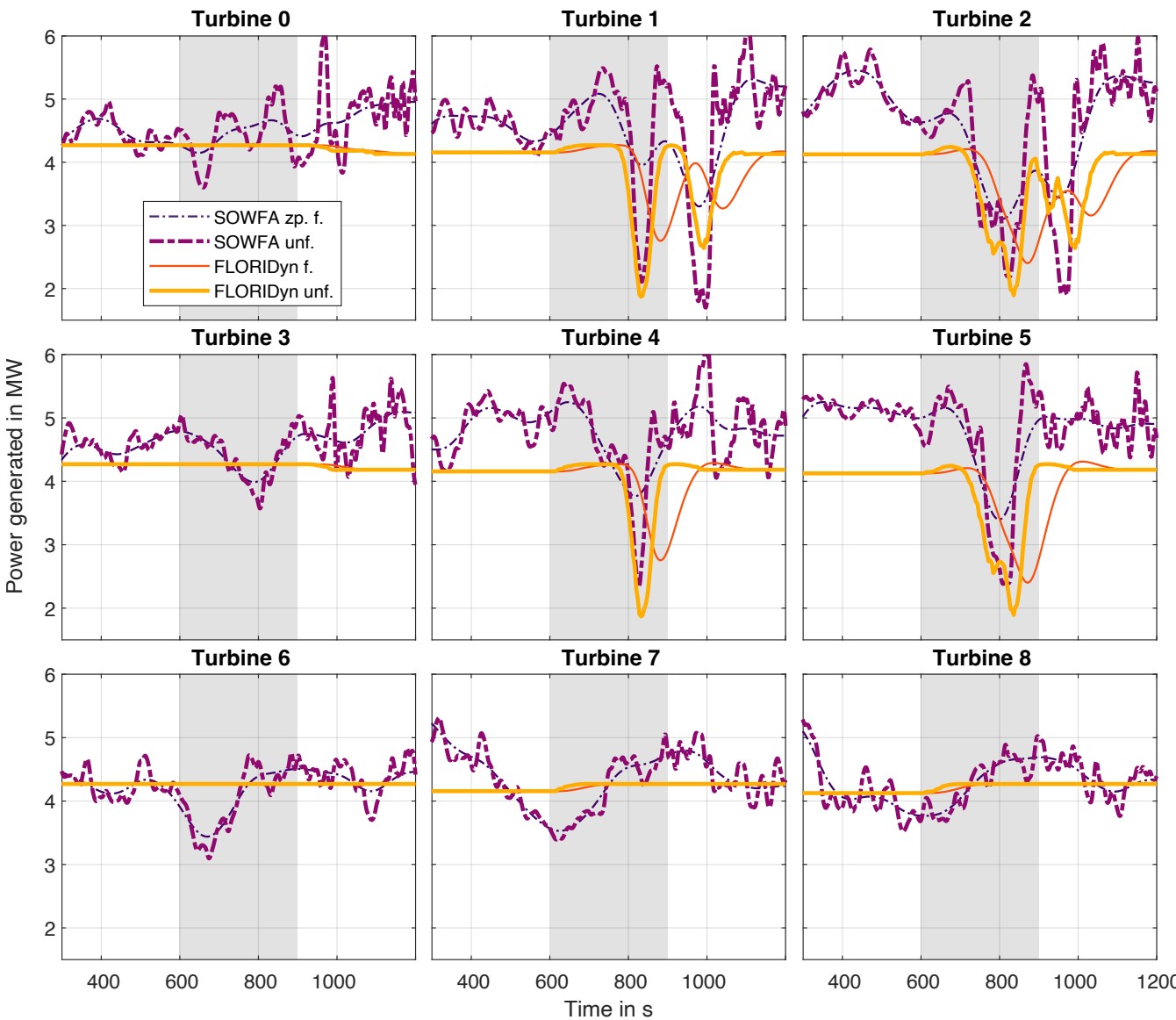

**Figure 11.** Power generated in the nine turbine case. The grey area marks the time window in which the wind direction linearly changes by $60°$. The plots are arranged to fit the layout of the wind farm in Figure 9. The data shows the zero-phase filtered (zp. f.) and the unfiltered (unf.) SOWFA data, as well as the filtered (f.) and unfiltered FLORIDyn data.

**Table 2.** Points in time at which the power generated in the nine turbine case is minimal due to wake interaction

| | Turbine 1 | | Turbine 2 | | Turbine 4 | Turbine 5 |
|---|---|---|---|---|---|---|
| | Min. 1 | Min. 2 | Min. 1 | Min. 2 | Min. 1 | Min. 1 |
| SOWFA (s) | 833.5 | 996.5 | 822 | 972 | 826.5 | 809 |
| FLORIDyn (s) | 832 | 992 | 836 | 992 | 832 | 836 |
| Error (s) | −1.5 | −4.5 | +14 | +20 | +5.5 | +27 |

data. Table 2 lists the points in time at which the power generated is minimal in SOWFA and in FLORIDyn, as well as the difference. This shows that FLORIDyn predicts the maximal wake influence $5.5\,\mathrm{s}$ later than in SOWFA. The filtering of the FLORIDyn data significantly worsens the quality of the result, in contrast to the simulations of the three turbine case. The filtering was applied on the data from the rotor plane. Thus, only modifying the way a turbine perceives the incoming, foreign wake in FLORIDyn will not improve the simulation for changing environments. As a result, future research has to improve the way a turbine dynamically influences its own wake. Turbine T1 shows similar behavior to T4: The generated power shows the wake influence of T3 first, but it also shows, after the wind direction stopped changing, the influence of the outskirts of the wake of T6. While the timing of this interaction shows good agreement, the magnitude of the interaction is considerably lower in FLORIDyn than in SOWFA. This could be due to a too fast recovering FLORIS wake, an inadequate wake superposition method or due to local turbulence levels, which FLORIDyn can not capture. T5 shows the overlapping influence of the wakes of T6 and T7. The two overlapping Gaussian influences do form a longer period of reduced generated power. This can be seen in both simulations. Table 2 shows the largest timing error between SOWFA and FLORIDyn for this wake influence. This could stem from an inaccurate wake interaction model and the way added turbulence is treated. T2 shows the most overlapping influences by the wakes of T3, T4, T6 and T7 in this order. While the first two overlapping interactions show good agreement, FLORIDyn shows poorer agreement with SOWFA for the influence of T6 and T7. Again, the SOWFA simulation suggests a larger decrease in generated power. A reason could be that the way wakes combine is not described accurately enough. The timing of the wake of T7 seems to be a bit too late as well. However, the SOWFA simulation recovers to its steady-state at about the same time as FLORIDyn. Additionally, all turbines with exception of T6 experience a small influence of the upstream turbines in the steady-state configuration. This effect is not noticeable in the SOWFA simulations. Concluding, FLORIDyn describes the timing of passing wake influences quite well. However, there are discrepancies in terms of magnitude and possibly in the way wakes combine their effects on downstream turbines.

### 3.3 Computational Performance

Table 3 contains the average computational time per time step, which is equivalent to $4\,\mathrm{s}$ simulation time. This can be compared to SOWFA, which can take around $5.8 \cdot 10^2\,\mathrm{s}$ to $5.4 \cdot 10^3\,\mathrm{s}$ per core, per time step, depending on the setup (van den Broek and van Wingerden, 2020). The FLORIDyn measurements were performed for two and three consecutive turbines and a $2 \times 2$ and $3 \times 3$ turbines wind farm. The times exclude plotting and the simulation setup time. A setup can take up to $3\,\mathrm{s}$, depending

**Table 3.** Computational performance

| Number of turbines | 2 | 3 | 4 | 9 |
|---|---|---|---|---|
| Total number of OPs | $2 \cdot 10^4$ | $3 \cdot 10^4$ | $4 \cdot 10^4$ | $9 \cdot 10^4$ |
| $t_{\text{comp. time per step}}$ (s) | $2.44 \cdot 10^{-2}$ | $5.87 \cdot 10^{-2}$ | $1.09 \cdot 10^{-1}$ | $6.13 \cdot 10^{-1}$ |

on how much data needs to be imported. The measurements were taken on a MacBook Pro (2019), 2.3 GHz 8-Core Intel i9 CPU, 32GB of 2667 MHz DDR 4 RAM, an SSD and MacOS Catalina (10.15.7). The simulation environment is Matlab 2020a without the use of toolboxes, such as the parallel computing toolbox, and without precompiled code, besides what is built into the simulation environment. These results naturally vary with the layout, atmospheric behavior, simulation settings, etc. and are only meant to give an estimation of the performance.

A first takeaway is that FLORIDyn simulates all cases faster than real-time: Within $4$ s, the simulation can perform between $164$ and $6.5$ simulation steps, depending on the number of turbines simulated. This results in $656$ $s_{\text{Sim}}$ in-simulation time for two turbines and goes down to $26$ $s_{\text{Sim}}$ for nine turbines in $4$ s of real-time. This opens up the needed computational headroom for a model based real-time control strategy and the necessary optimization. On the other hand, the times also do not offer a large time window for optimization. For instance in the three turbine yaw case, it takes roughly $300$ $s_{\text{Sim}}$ in simulated time until the yaw changes have propagated from the first turbine to the last turbine. To optimize the control actions with this model for the near future, parallel computing would be needed. With an increasing number of turbines, this time window decreases.

Generally, the computational time increases quadratically with $n_{\text{T}}^2 - n_{\text{T}}$, as $n_{\text{T}}$ turbines need to determine if they are in the wake of another turbine and calculate the influence. This growth in computational effort is assumed to decrease with larger wind farms: As the spacial dimension grows, not every turbine needs to consider all other turbines for interaction. Nevertheless, the simulation times will exceed what is practical for a wind farm with $\gg 3$ turbines.

There are multiple opportunities to improve performance which have not been utilized so far. The main aspect which increases computational effort is the interaction among the turbines. A first step can be to calculate the turbine interactions in parallel. A second step is to find a way to efficiently determine if a turbine is influenced by a wake. Furthermore, the number of OPs per turbine can be decreased and tuned: not all chains need to be equally long and OPs which wander out of the domain can be disregarded. Then, there is the fundamental question of whether the proposed structure of FLORIDyn can be improved, for instance by using less but more efficient OPs. Eventually, the programming platform can be switched to a choice which allows more specific optimization, e.g. C, C++ or Julia.

## 4 Conclusions and recommendations

In this paper, a new FLORIDyn model is presented and compared to SOWFA simulations. This model utilizes a Gaussian FLORIS model and concepts from the previously published FLORIDyn framework by Gebraad and van Wingerden (2014) to create a three-dimensional, dynamic and computationally lightweight wind farm model. The new FLORIDyn model is further capable of simulating its wakes under heterogeneous and time varying flow conditions. To achieve this, we presented a

mathematical approach to decouple wake and flow characteristics into two coordinate system which are connected by observation points. To simulate changing environmental conditions, a method to map sparse flow field measurements to a finer grid was presented which avoids the interpolation cost at runtime. The new FLORIDyn model shows good performance compared to SOWFA in terms of timing and is able to predict accurately when a downstream turbine is experiencing influences from upstream turbines.

Despite the considerable advancements over the old FLORIDyn implementation, there are still several aspect of the model which can be improved. The central aspect is how turbines influence wakes and how wakes are perceived by turbines. In this work we have decoupled the OP propagation speed from the effective wind speed, which effectively leads to a simpler, light-weight model while the wake behavior is still dynamically described. However, this way state changes reach downstream turbines too soon and in a sudden manner. Ideally this can be overcome by finding better, computational light weight methods to

model the influence of changing turbine states on the wake needs and also how a turbine reacts to dynamic changes in the flow. Another aspect that can be improved is related to the interface between FLORIDyn and FLORIS. FLORIS has been subject to many developments and improvements, and FLORIDyn can utilize these improvements if it improves the interface: With a generic interface, newer developments can be included and existing code can be used in a sustainable manner. The simulations also show that parameter tuning has to be more accessible and possibly needs to be performed online in some cases. The

next aspect which could be improved is the coupling of FLORIDyn with the turbulent environment of the real wind farm (or its surrogate). Combined with the changes to the OP propagation speed from this work, this can lead to a more uneven OP distribution with dense areas where high wind speeds decrease and sparse areas where low wind speeds increase. An extension to the model could feature a method to combine and generate OPs, depending on the density of OPs. Although, this could also lead to undesired information loss, depending on the implementation. To achieve better results, the wind field model has to be

replaced or enhanced by estimators. The latter would provide a more accurate estimate of wind speed, direction and ambient turbulence intensity for the FLORIDyn simulation. In the long term, a dynamic description of the environment could become part of the FLORIDyn model. This could also include effects like induction zones and speed-up between the turbines. The last aspect to consider for improvement is the performance. In its current implementation, FLORIDyn delivers its results at a low computational cost. This has to be maintained, if not improved to allow its use for dynamic real-time closed-loop control

algorithms in the future. The simulation also needs to be structurally improved to keep its low computational cost for wind farms with large numbers of turbines.

Concluding, the new FLORIDyn model is a promising concept with unique strengths. With FLORIS in its core, it utilizes an existing, successfully employed model and provides a new dimension in a challenging environment at a low computational cost. The model can already be adapted to work in a closed loop control design and shows more potential if the mentioned

aspects are improved.

*Code and data availability.* Both, the FLORIDyn and the SOWFA code are publicly available under the GPL-3.0 license. The FLORIDyn repository contains the entire Matlab code and the used measurements (power generated, blade pitch angle, rotor speed) from the used

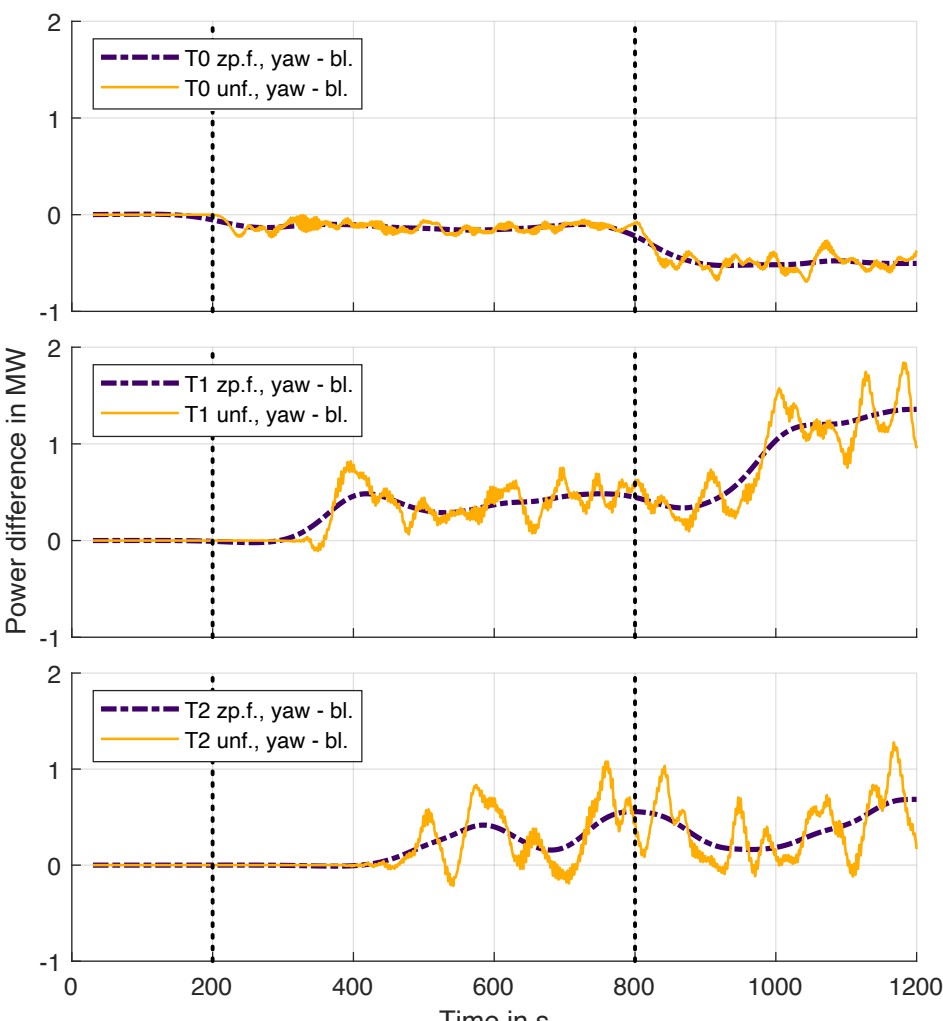

**Figure A1.** Difference between the yaw case in SOWFA and the baseline case with unfiltered and zero phase filtered data. Filtering was performed before calculating the difference. The dotted lines mark the start of the yaw angle changes of T0.

SOWFA simulations (Becker, 2022a). It is published by the Delft Center of Systems and Control (DCSC) . SOWFA is published by the National Renewable Energies Laboratory and written in C++ (National Renewable Energy Laboratory, 2020). The SOWFA files for the nine
turbine case are available to rerun and validate the claims from this paper (Becker, 2022b)

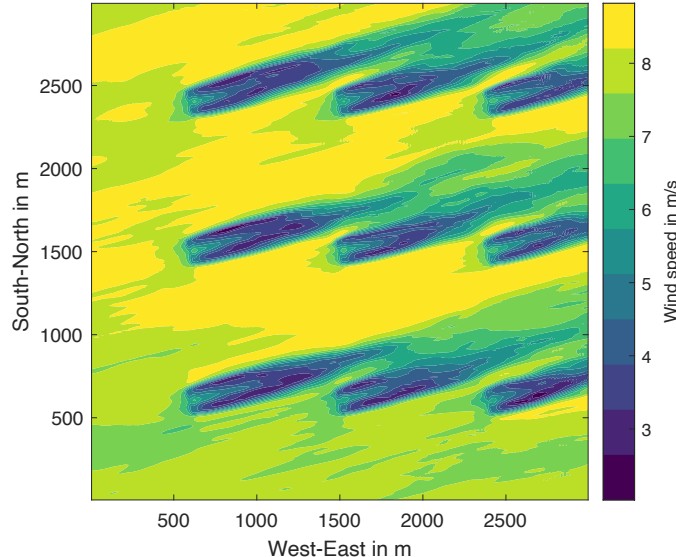

**Figure A2.** Averaged wind speed from $t = 500\,\mathrm{s}$ to $600\,\mathrm{s}$, divided into eleven speed sections.

## Appendix A: Additional plots and aspects of the simulation results

### A1  Unfiltered difference between yawed and baseline case

Figure A1 shows the difference between the power generated in SOWFA in the yaw case (Section 3.1.2) and the steady-state base line case (Section 3.1.1). Both simulations are performed in the same turbulent environment, something which would be impossible to achieve in realistic conditions. This way, the difference allows for a clearer interpretation of the influence of the yaw step, at least the timing. In comparison to Figure 8, Figure A1 shows the unfiltered data as well as the filtered one for all three turbines. T1 shows between $t = 312\,\mathrm{s}$ and $329\,\mathrm{s}$ a first reaction due to the changed wake of T0. T2 shows a first reaction between $t = 426\,\mathrm{s}$ and $442\,\mathrm{s}$. The filtered data shows a slightly earlier influence due to the nature of a zero phase filter.

### A2  Averaged velocity in the nine turbine case

Figure A2 shows the averaged wind speed in the nine turbine case from $t = 500\,\mathrm{s}$ to $600\,\mathrm{s}$ at hub height in SOWFA. 34 slices were used to average. The wind speed is binned into eleven wind speed sections which are plotted as contours. During the time of averaging the wind direction is constant. The average wind speed of the incoming air is at approximately $8\,\mathrm{ms}^{-1}$. However, between the turbine rows, the wind speed increases to a higher level, up to $9.44\,\mathrm{ms}^{-1}$ in some places. This could be explained by speed up effects: The turbines act as resistances in the flow field and the wind speed in the place of least resistance, between the turbines, increases. The effect has been observed and described in Bastankhah et al. (2021) as well for instance. Due to the speed up, the turbines further downstream experience higher wind speeds than the ones in free stream and generate more energy. Figure 11 quantifies the effect, where T2, T4 and T5 generate significantly more energy than T6 for instance. After the

wind direction change, the effect leads again T2 and T4 to generate more power. T1 is now in the situation T5 was in initially and also generates more energy. T5 however drops to a lower level. Without an added model for effects like these, FLORIDyn
(and FLORIS) will not be able to accurately describe the wind field.

*Author contributions.* MB developed and implemented the model under the supervision of and in discussion with BR, BD, DvdH, UK, DA and JWvW. The SOWFA simulations were setup by DvdH, BD provided the $C_T$ and $C_P$ look-up tables. JWvW was responsible for the funding. The manuscript was written by MB and corrected and proof-read by BR, BD, DvdH, UK, DA and JWvW.

*Competing interests.* The authors declare no competing interests.

*Acknowledgements.* This work is part of the research programme *Robust closed-loop wake steering for large densely space wind farms* with project number 17512, which is (partly) financed by the Dutch Research Council (NWO).

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
