# Peer review of "The revised FLORIDyn model: Implementation of heterogeneous flow and the Gaussian wake"

_Wind Energy Science, 2021_

## Referee Comment (RC2)

**Review of *The revised FLORIDyn model: Implementation of heterogeneous flow and the Gaussian wake* by M. Becker et al.**

Reviewer: M. Paul van der Laan, DTU Wind Energy

May 3, 2022

The authors propose several modifications to a dynamic engineering wake model and compare its results to large-eddy simulations for a small wind farm using three dynamics cases: a start-up case, a yaw-misalignment change case and wind direction change case.

The article is well written and contains detailed information about the proposed modifications. However, the description of the large-eddy simulations should be improved and the difference between the two models should be quantified instead of providing mostly qualitative statements. More detailed comments are listed below; they need to be addressed before the article can be considered for publication in Wind Energy Science.

**Main comments**

1. Introduction: It is nice that you have added a list of articles in the introduction. However, I do miss an early work on the Dynamic Wake Meandering model from Larsen et al. (2008) [1], which presents a concept (steady wake solution + turbulent meandering flow) that has been used by several authors that you refer to.

2. Section 2.1: While I am familiar with the near and far wake regions, I have not sure what you mean by the *potential core* (root vortex?), please clarify the physical meaning.

3. Section 2.2: You mention that overlapping wakes from multiple turbines are summed using a root-sum-square. I guess one could also use a different wake summation method here (as for example a linear sum)?

4. Section 2.3.1: You mention *The Gaussian FLORIS model does not have defined borders and it is three-dimensional.* I understand that you refer to the outer boundaries of the wake deficit. To be complete, you could add that the Gaussian wake model has a region near the turbine where the near wake is undefined (in PyWake [3] a constant value is used here based on the maximum deficit), which one could interpret as a border and a different wake region compared to the Gaussian wake region that represents the far wake.

5. Section 2.3.2: You write *An OP considers itself influenced by a foreign wake if the closest foreign OP is less than $\frac{1}{4}$ D away.* Where is the value of $\frac{1}{4}$ D based on?

6. Section 2.5: You mention the problem for using $C_T$ values higher than one. You could overcome this by using an alternative relation between $C_T$ and the axial induction, see for example eq. (2) and corresponding discussion of Madsen et al. (2019) [2].

7. Section 3.1: I lack information on the SOWFA simulation setup. What type of atmospheric inflow is applied? Is it a pressure driven neutral inflow, a conventionally neutral inflow (neutral at the surface but with a temperature inversion), are Coriolis forces applied? In addition, it is worthwhile to mention that the actuator line model is coupled to a blade element momentum method (or is this not the case?). You could possibly add a reference to

the numerical setup of SOWFA regarding the sub grid model and other important simulation details. The same questions apply to the nine turbine case presented in Section 3.2. Finally, you write *The wind direction is constant along the x axis.* I guess you refer to the *mean* wind direction, as the LES inflow will describe the distribution of inflow wind directions. This may seem pedantic, but it is an important detail.

8. Section 3.2: How is 60° inflow wind direction change realized in SOWFA? This is not a trivial setup in CFD as it could violate mass conservation. Could it be that you employ a setup similar to Stieren et al. (2021) [4]? Please clarify.

9. Do you use the same yaw controller in both the SOWFA and FLORIDyn simulations in the yaw angle and wind direction change studies? If this is not the case, then this could be a source of the observed differences.

10. Section 2.2: In the discussion you mention several causes for the differences between SOWFA and FLORIDyn. I think you could also mention that the wake superposition method could be investigated, as this is normally a major source of error (at least for steady-sate wake models). In addition, I lack a quantitative analysis, as all statements about the comparisons are qualitative.

11. I would change the title of Section 3.3 (Performance) to Computational Performance or something that makes it clear that you investigate the computational effort and not the performance in terms of accuracy.

12. I lack a sentence in the conclusion regarding the change of using freestream wind speed as travel wind speed instead of the local wind speed, and how this has led to simpler but also a too fast reacting model.

**Minor comments**

1. Line 206: *wolrd* should be *world*.

2. Line 225: You write *actuator disc theory*, I would refer to this as *1D momentum theory*, but I do understand that both terms can be used.

**References**

[1] Larsen, G. C., Madsen, H. A., Thomsen, K., and Larsen, T. J. Wake meandering: a pragmatic approach. *Wind Energy*, 11(4):377–395, 2008.

[2] Madsen, H. A., Larsen, T. J., Pirrung, G. R., Li, A., and Zahle, F. Implementation of the blade element momentum model on a polar grid and its aeroelastic load impact. *Wind Energy Science*, 5(1):1–27, 2020.

[3] Pedersen, M. M., van der Laan, P., Friis-Møller, M., Rinker, J., and Réthoré., P. DTUWindEnergy/PyWake: PyWake, 2019.

[4] Stieren, A., Gadde, S. N., and Stevens, R. J. Modeling dynamic wind direction changes in large eddy simulations of wind farms. *Renewable Energy*, 170:1342–1352, 2021.

---

## Author Comment (AC1)

**Response to the reviews of *The revised FLORIDyn model: Implementation of heterogeneous flow and the Gaussian wake**

Marcus Becker

June 10, 2022

**Abstract**

I would like to thank both reviewers, Jamie Liew and M. Paul van der Laan, for their in-depth review of the presented paper. Their questions have shown where the script has shortcomings and fails to explain the presented research adequately. I present the changes made to the paper in this document.

The major comments of both reviewers are sorted chronologically by section and topic. In some cases, both reviewers commented on the same aspect. In these cases only one response was written, which aims to address both comments. In the following text, Jamie Liew will be referred to as "Reviewer 1" and M. Paul van der Laan as "Reviewer 2".

Correspondence: *marcus.becker@tudelft.nl*

**Section 1, Introduction**

**Dynamic Wake Meandering (DWM) model**

**Reviewer 1**   Section 1: I think it is worth mentioning literature on the Dynamic Wake Meandering (DWM) model as it has been verified and used extensively in dynamic wind farm simulations in HAWC2, FAST.Farm and HAWC2Farm. The DWM model shares qualities with the presented FLORIDyn approach, particularly in the use of observation points (similar to passive tracers in DWM literature), and in the way it extends a static wake model to behave dynamically. See: Larsen, G. C., Madsen, H. A., Thomsen, K., & Larsen, T. J. (2008). Wake meandering: A pragmatic approach. Wind Energy, 11(4), 377–395. https://doi.org/10.1002/we.267, and Madsen, H. A., Larsen, G. C., Larsen, T. J., Troldborg, N., & Mikkelsen, R. (2010). Calibration and validation of the dynamic wake meandering model for implementation in an aeroelastic code. Journal of Solar Energy Engineering, Transactions of the ASME, 132(4), 1–14. https://doi.org/10.1115/1.4002555, etc

**Reviewer 2**   However, I do miss an early work on the Dynamic Wake Meandering model from Larsen et al. (2008) [1], which presents a concept (steady wake solution + turbulent meandering flow) that has been used by several authors that you refer to.

**Response**   Thank you for the comments, I have added a paragraph about the DWM model in the Introduction:
*The Dynamic Wake Meandering (DWM) Model, first presented by Larsen et al. (2008) and later calibrated and refined by Madsen et al. (2010), proposes an approach much closer to established CFD methods. The model follows a pseudo-Lagrangian approach and creates turbulence boxes around the wake deficit which is created by the turbine. These boxes are then subject to a synthetic turbulent wind field, which allows the modeling of the wake meandering effect. The DWM model puts a focus on load estimation next to the power generated and simulates the turbine by coupling a CFD actuator disc model with an aeroelastic model. Compared to the other mentioned models, the DWM model presents a synergy of CFD methods with engineering approaches.*

**Code**

**Reviewer 1**   Line 85 - The code which is cited here is a direct link to the GitHub page. For longevity, it is advised to register the code with an open-access repository with a DOI (such as Zenodo) and to cite the newly minted DOI.

**Response**   Thank you for noticing that! I have uploaded the code to the Dutch dataset publication platform data.4tu.nl where it has received the DOI 10.4121/19867846 and has just been published. The citations in the text are adapted accordingly.

**Section 2: A new parametric dynamic wind farm model**

**Cp & Ct**

**Reviewer 1**   Line 98 - Please define the notation CP and CT.

**Response**   Thank you for noting the missing variable introduction.
   *To get $C_P$ and $C_T$ values closer to the validation platform SOWFA, a lookup table was generated (Section 2.5).*
was changed to
   *To get the power coefficient ($C_P$) and the thrust coefficient ($C_T$) values closer to the validation platform SOWFA, a lookup table was generated (Section 2.5).*

**Coordinate System**

**Reviewer 1**   Section 2.1 - Please define the coordinate system used (x = longitudinal, y = lateral, z = vertical).

**Response**   Added a paragraph at the beginning of section 2:

*In the wake coordinate system, $\mathcal{K}_1$, $x_1$ describes the downwind direction, $y_1$ the horizontal crosswind direction and $z_1$ the vertical crosswind direction, see Figure 3. In this coordinate frame, the rotor center is always located at $[0,0,0]^\top$. This coordinate system is not to be confused with the longitudinal ($x_0$), latitudinal ($y_0$) and vertical ($z_0$) world coordinate system $\mathcal{K}_0$. Their relation is described in Section 2.4.*

And adapted Figure 1 to include $x_1$ and $y_1$.

**Potential Core**

**Reviewer 2**   Section 2.1: While I am familiar with the near and far wake regions, I have not sure what you mean by the potential core (root vortex?), please clarify the physical meaning.

**Response**   The potential core is a concept which comes from jets in a coflow and describes a triangular area with constant speed behind the outlet. This concept is here adapted to approximate the near field behaviour.

Added a sentence in Section 2.1:

*The potential core is a region from jets in a coflow (Lee and Chu, 2003). Here, it is used to approximate the immediate region behind the rotor plane.*

Added source: Lee, J. H. W. and Chu, V. H.: Turbulent Round Jet in Coflow, in: Turbulent Jets and Plumes, pp. 179–209, Springer US, Boston, MA, https://doi.org/10.1007/978-1-4615-0407-8_6, http://link.springer.com/10.1007/978-1-4615-0407-8_6, 2003.

**Wake interaction model**

**Reviewer 2**   Section 2.2: You mention that overlapping wakes from multiple turbines are summed using a root-sum-square. I guess one could also use a different wake summation method here (as for example a linear sum)?

**Response**   Yes indeed, in this case we chose to follow the original FLORIDyn approach, but other options are possible.

Changed

*The resulting reduction of the free wind speed is calculated as follows: Equation 1 where $u_{free,OP}$ is the free wind speed at the OP's location.*

to

*In this model, the resulting reduction of the free wind speed is calculated as follows: Equation 1 where $u_{free,OP}$ is the free wind speed at the OP's location. This wake interaction model could also be exchanged for another formulation.*

**Wake borders**

**Reviewer 2**   Section 2.3.1: You mention The Gaussian FLORIS model does not have defined borders and it is three-dimensional. I understand that you refer to the outer boundaries of the wake deficit. To be complete, you could add that the Gaussian wake model has a region near the turbine where the near wake is undefined (in PyWake [3] a constant value is used here based on the maximum deficit), which one could interpret as a border and a different wake region compared to the Gaussian wake region that represents the far wake. .

**Response**   The wake formulation by Bastankhah and Porté-Agel does adapt the model of a jet in a coflow to the turbine wake, as discussed earlier in this response. This also features a near wake description which consists of the potential core and a Gaussian shaped transition from the core to the free flow. I would therefore argue that the wake is fully defined from the rotor plane on-wards.

No changes have been made to the text.

Bastankhah, M. and Porté-Agel, F.: Experimental and theoretical study of wind turbine wakes in yawed conditions, Journal of Fluid Mechanics, 806, 506–541, https://doi.org/10.1017/jfm.2016.595, 2016

**Deflection**

**Reviewer 1** Equation (2) and (3) - $\delta$ defines only a lateral deflection - could it be useful to define a vertical deflection? For example, when applying the helix approach?

**Response** Yes, indeed, the formulation can be easily extended to other centerline deflection models which also take tilt into account. As future work it could also be possible to design a deflection model coupled to the real world coordinates to take the topography into account.
Added two sentences after Eq. (2) and (3):

*Note that this model only assumes a horizontal deflection. To add a vertical deflection, due to rotor tilt for instance, Equation (3) needs to be adapted accordingly.*

**Interaction**

**Reviewer 1** Line 171 - Is there a particular reason for choosing 1/4D?

**Reviewer 2** Section 2.3.2: You write An OP considers itself influenced by a foreign wake if the closest foreign OP is less than 1 4 D away. Where is the value of 1 4 D based on?

**Response** There is no particular reason for the value, I have extended the passage and added a brief discussion:

*This is an arbitrary chosen threshold to reduce the number of OPs for the interaction interpolation. As the outer wake OPs represent the most recovered sections of the wake, this still results in a smooth influence transition.*

**Propagation Speed**

**Reviewer 1** 180 to 189 - The authors mention the use of the mean ambient wind speed to propagate the OPs. This is a fair compromise and is supported by some literature. There is, however, other literature that provides different conclusions. See, for example, Andersen, S. J., Sørensen, J. N., & Mikkelsen, R. F. (2017). Turbulence and entrainment length scales in large wind farms. Philosophical Transactions of the Royal Society A: Mathematical, Physical and Engineering Sciences, 375(2091). https://doi.org/10.1098/rsta.2016.0107, Which measures a propagation speed of around 0.69 to 0.88 times $U_\infty$.

**Response** Added a sentence at the end of Section 2.3.3 with the suggested source:

*In future work, the wake propagation speed could be a tuning parameter which is set depending on atmospheric conditions such as the turbulence intensity for instance (Andersen et al., 2017).*

**OP propagation step translation wake to world**

**Reviewer 1** Equation 9 - This formulation assumes that the wind direction is uniform across space and is known a priori. Is this applicable to wind fields that have spatially non-uniform wind direction changes? What about wind fields where the wind direction is not clearly defined, or is gust-like?

**Response** I respectfully disagree with the statement as Eq.(9) uses $\varphi_{0,OP}$ which refers to the wind speed at an OP's specific location. Contrary, Eq.(7) explains the transformation for a uniform wind direction $\varphi$. A sentence was added for clarification:

*Note that $\varphi_{0,OP}$ refers to one OP's individual wind direction, other OPs may have different values. This means that each OP propagates on its own and non-uniform wind directions can be simulated.*

**Blade Pitch-TSR Look-Up Table**

**Reviewer 1** Line 231 - The authors use a blade pitch-TSR look-up table. These tables are typically generated using static turbine simulations. Is it reasonable to use static lookup tables in a dynamic simulation? What are the implications of doing so?

**Response**  This is indeed a weakness of using the proposed tables, and it is generally an issue as we are moving to more dynamic control methods. I have added a discussion in the first paragraph of Section 2.5:

*It has to be added that these tables are generated from a grid of high fidelity simulations, where the coefficients were read after the simulation converged to a steady state. The tables therefore only approximate the effect a changing turbine state and changing wind field conditions. Control approaches for axial-induction-based controllers, such as the one presented by Annoni et al., 2016, successfully use similar look-up tables, which is why we assume these to be sufficient. Nevertheless, an extension for dynamic circumstances would be a valuable addition for future work, but is also connected to a significant computational effort.*

**Reviewer 1**  Line 238 - Have the authors considered using an alternative relationship between $C_T$ and $a$? For example, Aagaard Madsen, H., Juul Larsen, T., Raimund Pirrung, G., Li, A., & Zahle, F. (2020). Implementation of the blade element momentum model on a polar grid and its aeroelastic load impact. Wind Energy Science, 5(1), 1–27. https://doi.org/10.5194/wes-5-1-2020, which uses a cubic equation ($a = k_3 C_T^3 + k_2 C_T^2 + k_1 C_T + k_0$), which extends into the high thrust region. It can also be inverted analytically using a cosh substitution.

**Reviewer 2**  Section 2.5: You mention the problem for using CT values higher than one. You could overcome this by using an alternative relation between CT and the axial induction, see for example eq. (2) and corresponding discussion of Madsen et al. (2019) [2].

**Response**  We have considered it, but as the value of $C_T$ is limited it did not seem necessary. However, it is good to link to the proposed source for future improvements:

*In future work this expression could be substituted, for instance by the polynomial approximation of Madsen et al. (2020). It extends $a(C_T)$ to $C_T$ values above 1. However, as $C_T$ is limited in this work, this extension is not necessary.*

**Reviewer 1**  Line 245 - The authors mention the use of a constant value of the power-yaw exponent $p_p$. It has been shown in literature and measurements that Pp can vary, especially for turbines in wake conditions. See: Liew, J., Urbán, A. M., & Andersen, S. J. (2020). Analytical model for the power-yaw sensitivity of wind turbines operating in full wake. 1–18. https://doi.org/10.5194/wes-2019-65 and Howland, M. F., González, C. M., Martínez, J. J. P., Quesada, J. B., Larrañaga, F. P., Yadav, N. K., Chawla, J. S., & Dabiri, J. O. (2020). Influence of atmospheric conditions on the power production of utility-scale wind turbines in yaw misalignment. Journal of Renewable and Sustainable Energy, 12(6). https://doi.org/10.1063/5.0023746

Similar effects are expected for how $C_T$ varies with yaw angle. As the presented model is intended for control purposes, it is important to consider how the power loss is modeled for yaw steering in future studies. Is a variable $p_p$ a possibility in FLORIDyn?

**Response**  A variable $p_p$ is not implemented in the current model but is definitely an option. In future work these implementations should be extended to be more accurate. I have added the following note:

*For simplicity's sake we assume $p_p$ to be a constant value. This could be extended by the work presented by Liew et al. (2019), which takes the presence of other wakes into account. Similarly, Howland et al. (2020), presents an adaptation for locally varying wind profiles.*

**Section 3: Simulation Results**

**Simulation Setup**

**Reviewer 2**  Section 3.1: I lack information on the SOWFA simulation setup. What type of atmospheric inflow is applied? Is it a pressure driven neutral inflow, a conventionally neutral inflow (neutral at the surface but with a temperature inversion), are Coriolis forces applied? In addition, it is worthwhile to mention that the actuator line model is coupled to a blade element momentum method (or is this not the case?). You could possibly add a reference to the numerical setup of SOWFA regarding

the sub grid model and other important simulation details. The same questions apply to the nine turbine case presented in Section 3.2. Finally, you write The wind direction is constant along the x axis. I guess you refer to the mean wind direction, as the LES inflow will describe the distribution of inflow wind directions. This may seem pedantic, but it is an important detail.

**Response**  I do understand your questions and acknowledge that the description of the SOWFA simulations is brief. As a reaction, I have made multiple changes to the script. The introduction of Section 3 now contains the sentence
   *The inflow boundary conditions for SOWFA are provided by a precursor simulation which simulates a horizontally homogenous, conventionally neutral atmospheric boundary layer including Coriolis effects.*

Furthermore, we have published the in- and output files of the nine turbine case under the DOI 10.4121/20026406. A link as been added as a citation in Section 3.2.
The phrase
   *The wind direction is constant along the x axis.*
was changed to
   *The mean wind direction is constant along the x axis.*

Regarding your question about the actuator line method, we are not using a coupling to an aeroelastic solver like FAST, but rather the standard implementation in SOWFA. I have adapted the phrasing from
   *In SOWFA, the turbines are modelled with the Actuator Line Method (ALM)*
to
   *In SOWFA, the turbines are modelled with the built-in Actuator Line Method (ALM) implementation*

**Filtering**

**Reviewer 1**  Figure 6 - The use of a low pass filter on the power signal is justified to replicate the delay in the turbine response. However, what is the motivation for using a zero-phase low pass filter on one signal and a causal filter on the other? Even with the same damping and cut-off frequency, I expect there to be a noticeable phase delay in the causal filter compared to the non-causal filter (which I believe is visible in Figure 11). So comparing the signals from different filters may be misleading. Would it make sense to use the same filter type on both the SOWFA and FLORIDyn results?

   Continuing from the previous point, have the authors considered using a causal filter in real-time instead of in post-processing? From the results, it appears that FLORIDyn responds quite quickly to changes in the wind field. In reality, there will be some delay due to the inertia in the turbine rotor.

**Response**  I understand the confusion and it is something that I have struggled with to explain properly during writing. The two filters serve two different purposes:

1. The SOWFA zero phase filter aims to reduce the visual effect of turbulences on the plots. The resulting filter is indeed non-causal but does not add a phase shift to the data and hopefully allows for a better interpretation of the simulation data. Especially the data from Figure 7 (three turbine case with yaw angle change) is very noisy and hard to interpret if no filter is applied.

2. The FLORIDyn low-pass filter on the other hand serves another purpose, as it is meant to visualize what would happen if the input to the model would be lowpass filtered. It is causal and therefore offers a valid option to be implemented. It poses the question: Can we "fix" the steep edges in FLORIDyn by "just" lowpass filtering whatever comes as input at the rotor plane? In the three turbine case, this seems to be the case, but in the nine turbine case the same filter significantly worsens the results due to the phase shift it adds.

   I also believe that it is valid to compare the outputs of both filters as the comparison poses the question: "Could a low-pass filtered FLORIDyn signal describe the underlying wake dynamics of the validation simulation?" You could also argue that the SOWFA signal is already low-pass filtered as

SOWFA models the rotor/generator response to the changes in the flow field.

Regarding the question if we have considered using the filter at run-time, yes that could be done, but it would not change the way the simulation runs, at least not, if applied to the power generated as it is done here.

Following your questions, I have made multiple changes to the text. In the paragraph about the zero-phase SOWFA filter I have added the phrase:

*This non-causal filter is added to aid the visual interpretation of the simulation results.*

The paragraph about the FLORIDyn filtering has been changed from

*A regular second-order low-pass filter with the same d and ω is used for the FLORIDyn data. This filter means to visualize a possibly smoother interaction of turbines in FLORIDyn with their environment or vice versa. Due to the changes made to the OP travel speed in Section 2.3.3, the wake interaction in FLORIDyn can be very abrupt, as will be discussed in Section 3.1.1. The presented filter could be used at runtime in FLORIDyn and could be interpreted as a low frequency response of the turbine to sudden environmental changes.*

to

*A regular second-order low-pass filter with the same d and ω is used for the FLORIDyn data. This causal filter visualizes how low-pass filtering would effect the predicted power generated. This could have advantages due to the changes made to the OP travel speed in Section 2.3.3 which can lead to a very abrupt wake interaction, as will be discussed in Section 3.1.1. However, the filter also naturally adds a phase shift to the data, an effect which might not be desired.*

Another note has been added, following the filter description:

*Note that the two filters have different purposes: The non-causal SOWFA filter aims to help to interpret the simulation results, while the causal FLORIDyn filter explores if and when the use of a low-pass filter would be advantageous or if it would decrease the quality of the results.*

**Changing wind direction**

**Reviewer 1**  Section 3.2 - The nine turbine case is an interesting showcase of FLORIDyn's ability to respond to a wind direction change. The process of simulating a wind direction change is nuanced and often requires many physics-defying assumptions. It is worth mentioning these assumptions, for example: The uniformly rotating wind field in the FLORIDyn simulations is a fair compromise to showcase the model and to test controllers. Could you please elaborate on how the wind direction change is performed in SOWFA? It is unclear if the wind field is also rotated uniformly in SOWFA, which will break continuity, or it is using a different method such as Stieren, A., Gadde, S. N., & Stevens, R. J. A. M. (2021). Modeling dynamic wind direction changes in large-eddy simulations of wind farms. Renewable Energy, 170, 1342–1352. https://doi.org/10.1016/j.renene.2021.02.018 and Andersen, S. J., S rensen, N. N., & Kelly, M. (2021). Les modelling of highly transient wind speed ramps in wind farms. Journal of Physics: Conference Series, 1934(1). https://doi.org/10.1088/1742-6596/1934/1/012015

Continuing from the previous point, Stieren shows noticeable hysteresis effects on the power output of turbines during a wind direction change. It is unclear if the SOWFA simulation is set up in a way to capture these effects.

**Reviewer 2**  Section 3.2: How is 60° inflow wind direction change realized in SOWFA? This is not a trivial setup in CFD as it could violate mass conservation. Could it be that you employ a setup similar to Stieren et al. (2021) [4]? Please clarify.

**Response**  We achieve the wind direction change by using SOWFA's build-in utilities during the precursor simulation. As SOWFA is a well established LES solver that has been used and validated many times, so we trust that the SOWFA simulation is of high-fidelity and comparable to other state of the art LES solvers. We also acknowledge that the presented case is artificial and simplifies the behavior in a real world environment. Regarding the hysteresis effect, we believe that it should have a minor effect on the presented simulation, as we simulate a linear gradient and Stieren et al. simulates

a sinusoidal variations in the wind direction.

I have made multiple additions to the script to address your points:

The introduction of Section 3.2 now contains the note:

*The change in wind direction is achieved by using SOWFA's built-in utility to specify the wind speed and wind direction at a certain height and time.*

The paragraph further concludes with:

*For more information see the dataset which contains the SOWFA files for the case and the precursor simulation (Becker2022b).*

The source contains the files used to simulate the case and precursor. It also contains the outputs, besides the flow field snapshots. We hope that this provides sufficient information for interested readers and other researchers who aim to validate / compare the study. The dataset publication has the DOI 10.4121/20026406.

**Yawing during the wind direction change**

**Reviewer 1**   Both SOWFA and FLORIDyn simulations set the turbine yaw angle to match the changing wind direction. This is reasonable as a show-case of the flow model, however, it is idealistic and does not reflect the delay that a turbine experiences when changing its orientation.

**Reviewer 2**   Do you use the same yaw controller in both the SOWFA and FLORIDyn simulations in the yaw angle and wind direction change studies? If this is not the case, then this could be a source of the observed differences.

**Response**   We do not use a closed-loop yaw controller but apply a prescribed motion to the turbines during the wind direction change. This eliminates another source of uncertainty and complexity. As the wind direction is changing with $0.2° s^{-1}$ it is reasonable to assume that a real turbine could follow. In the text I changed

*During the simulation, the turbines maintain a yaw angle of $0°$.*

into

*During the simulation, the turbines maintain a yaw angle of $0°$ and turn with the wind. For simplicity we assume ideal wind direction tracking capabilities and apply a prescribed motion.*

**Differences between SOWFA and FLORIDyn**

**Reviewer 2**   *Note by the author: I assume that the reviewer meant to refer to Section 3.2 due to the comment and the place in the otherwise chronological review.*

Section 2.2: In the discussion you mention several causes for the differences between SOWFA and FLORIDyn. I think you could also mention that the wake superposition method could be investigated, as this is normally a major source of error (at least for steady-sate wake models). In addition, I lack a quantitative analysis, as all statements about the comparisons are qualitative.

**Response**   Thank you for the addition, I have added the wake superposition method as a possible source of error in section 3.2. The part

*While the timing of this interaction shows good agreement, the magnitude of the interaction is considerably lower in FLORIDyn than in SOWFA. This could be due to a too fast recovering FLORIS wake, a stronger than expected influence of a passing wake on the turbine or also due to local turbulence levels, which FLORIDyn can not capture.*

has been adapted to

*While the timing of this interaction shows good agreement, the magnitude of the interaction is considerably lower in FLORIDyn than in SOWFA. This could be due to a too fast recovering FLORIS wake, an inadequate wake superposition method or due to local turbulence levels, which FLORIDyn can not capture.*

Regarding the lack of quantitative statements, I have added Table 1 (in the paper Table 2) with

Table 1: Points in time at which the power generated in the nine turbine case is minimal due to wake interaction

|  | Turbine 1 | | Turbine 2 | | Turbine 4 | Turbine 5 |
|---|---|---|---|---|---|---|
|  | Min. 1 | Min. 2 | Min. 1 | Min. 2 | Min. 1 | Min. 1 |
| SOWFA (s) | 833.5 | 996.5 | 822 | 972 | 826.5 | 809 |
| FLORIDyn (s) | 832 | 992 | 836 | 992 | 832 | 836 |
| Error (s) | $-1.5$ | $-4.5$ | $+14$ | $+20$ | $+5.5$ | $+27$ |

the points in time at which the power generated is minimal due to wake influence. This table supports the claim that the timing of FLORIDyn shows good agreement with the SOWFA simulation, also considering the time scales of the dynamic effects.

In the discussion of the power generated of Turbine 4 I have added the phrases:

*Table 2 lists the points in time at which the power generated is minimal in SOWFA and in FLORIDyn, as well as the difference. This shows that FLORIDyn predicts the maximal wake influence 5.5 s later than in SOWFA.*

Another phrase was added following the discussion of Turbine 5:

*Table 2 shows the largest timing error between SOWFA and FLORIDyn for this wake influence. This could stem from an inaccurate wake interaction model and the way added turbulence is treated.*

As for the other differences, such as power generated or wake width, I feel the model is too unrefined yet and the differences too high for quantitative statements to be useful. As mentioned in the paper, only rough parameter tuning has been part of the work, also the validation simulation shows room for improvement. Generally, I see this work as an advancement for the dynamic modeling of wakes, but still at an early stage.

**Title of the Section Performance**

**Reviewer 2**   I would change the title of Section 3.3 (Performance) to Computational Performance or something that makes it clear that you investigate the computational effort and not the performance in terms of accuracy.

**Response**   The title has been changed to *Computational Performance.*

**Simulation step duration**

**Reviewer 1**   Line 397 - I don't understand the phrasing "from 164 to 6.5 simulation steps". Perhaps rephrase?

**Response**   The statement has been rephrased from
*Within 4 s, the simulation can perform from 164 to 6.5 simulation steps, which results in 656 $s_{Sim}$ for two turbines to 26 $s_{Sim}$ for nine turbines.*
to
*Within 4 s, the simulation can perform between 164 and 6.5 simulation steps, depending on the number of turbines simulated. This results in 656 $s_{Sim}$ for two turbines and goes down to 26 $s_{Sim}$ for nine turbines in 4 s of real time.*

**Single- or Multi-core**

**Reviewer 1**   Line 401 - Does the presented FLORIDyn simulation run on a single core? From my understanding, MATLAB will use parallel (or at least multithreaded) computing for certain operations by default.

**Response**   That might be case and it was not my intention to possibly hide some code execution details. I have adapted the phrase

*The simulation environment is Matlab 2020a without the use of toolboxes, precompiled code or parallel computing.*

to

*The simulation environment is Matlab 2020a without the use of toolboxes, such as the parallel computing toolbox, and without precompiled code, besides what is built into the simulation environment.*

to stronger emphasize that the implementation itself does not add parallelization.

**Computational time**

**Reviewer 1**   Table 2 - The computational time is quite impressive for so many OPs. To strengthen the argument, it would be nice to see a comparison with the iteration time for SOWFA. Additionally, it may be worth mentioning the required computational resources for both (CPU seconds per iteration, memory, etc).

**Response**   That is a good point, however, I feel like it is difficult to put absolute numbers next to the FLORIDyn numbers as the computational time in SOWFA strongly depends on the settings used (ALM/ADM, grid resolution, domain size, etc.). I have added the following phrase:

*This can be compared to SOWFA, which can take around $5.8 \cdot 10^2$ s to $5.4 \cdot 10^3$ s per core, per time step, depending on the simulation setup (van den Broek and van Wingerden, 2020).*

Added Source:

van den Broek, M. J. and van Wingerden, J. W.: Dynamic Flow Modelling for Model-Predictive Wind Farm Control, Journal of Physics: Conference Series, 1618, 022 023, https://doi.org/10.1088/1742-6596/1618/2/022023, 2020.

**Reviewer 1**   Line 403 - Does the simulation time increase exponentially? I would expect it to increase linearly with the number of turbines, or perhaps quadratically if the OPs interact with each other.

**Response**   You are right, thank you for spotting that mistake. As each turbine needs to determine if it is influenced by another wake, the computational effort increases quadratically.

The phrase has been changed from

*Generally, the computational effort increases exponentially with the number of turbines.*

to

*Generally, the computational time increases quadratically with $n_T^2 - n_T$, as $n_T$ turbines need to determine if they are in the wake of another turbine and calculate the influence.*

**Section 4: Conclusions and recommendations**

**Use of freestream / ambient wind speed**

**Reviewer 1**   Line 431 - The authors mention adapting FLORIDyn for turbulent conditions. As the presented model propagates the OPs using the ambient wind speed, I am curious how FLORIDyn can be adapted for turbulent flows. Will the OPs propagate based on a turbulent wind field? Will there be issues with the untidy movement of so many OPs? How can the issues outlined in Section 2.3.3 be resolved?

**Response**   With this implementation OPs are following the same background flow, so I don't expect them to necessarily "take over" other OPs, but the implementation could indeed lead to areas with a high density and low density of OPs. This has added as a brief discussion:

The sentence

*The next aspect which could be improved is the coupling of FLORIDyn with the turbulent environment of the real wind farm (or its surrogate).*

was extended by

*The next aspect which could be improved is the coupling of FLORIDyn with the turbulent environment of the real wind farm (or its surrogate). Combined with the changes to the OP propagation speed from this work, this can lead to a more uneven OP distribution with dense areas where high wind speeds decrease and sparse areas where low wind speeds increase. An extension to the model could feature a method to combine and generate OPs, depending on the density of OPs. Although, this could also lead to undesired information loss, depending on the implementation.*

**Reviewer 2**  I lack a sentence in the conclusion regarding the change of using freestream wind speed as travel wind speed instead of the local wind speed, and how this has led to simpler but also a too fast reacting model.

**Response**  I tried to comment on that in the sentences

*The central aspect is how turbines influence wakes and how wakes are perceived by turbines: while the wake behaviour is dynamically described, the influence of changing turbine states on the wake needs a better dynamic description. Also how a turbine reacts to dynamic changes in the flow might require a better approximation.*

But I do see that this does not comment on the issue sufficiently. I have extended it to the following:

*The central aspect is how turbines influence wakes and how wakes are perceived by turbines. In this work we have decoupled the OP propagation speed from the effective wind speed, which effectively leads to a simpler, light-weight model while the wake behaviour is still dynamically described. However, this way state changes reach downstream turbines too soon and in a sudden manner. Ideally this can be overcome by finding better, computational light weight methods to model the influence of changing turbine states on the wake needs and also how a turbine reacts to dynamic changes in the flow.*

**Control loop**

**Reviewer 1**  Line 437 - The authors suggest using FLORIDyn within the control loop. Out of curiosity, could you elaborate on how such a setup would work? How would measured observations be translated into a state in a FLORIDyn simulation?

**Response**  I would imagine a classic model predictive control framework, where, given the current state of the simulation, an optimization is performed over the near future. Given the low computational cost and a manner to parallelize the search, multiple versions of FLORIDyn could test different yaw angles / axial induction factors over a given time horizon to minimize a cost function. There are certainly challenges to overcome until this can be achieved, but I believe it can be done.

Regarding the measured observations to states: The presented wind field model already assumes distributed, scattered measurements of the quantities of interest. Met. masts could contribute to that or turbine sensors and estimators. The latter is also discussed in the just published work on the FLORIDyn framework at Torque, in case you are interested: https://iopscience.iop.org/article/10.1088/1742-6596/2265/3/032103

I have also looked at the sensitivity of that implementation at the PhD Seminar in Porto, If you recall that. The work was published in another way by Vinit Dighe at Torque: https://iopscience.iop.org/article/10.1088/1742-6596/2265/2/022062/meta. Here he used the surrogate model to do parameter tuning.

I'd be happy to discuss this as well at a fitting occasion, as closing the loop will be the big challenge to prove that dynamic model based wind farm control

As I understand the comment as a personal interest, I have not adapted the text.

**Typographical corrections and minor comments**

These corrections are by Jamie Liew if not marked otherwise.

- Stay consistent with the use of American or British English. For example (if you are sticking to American English):

- maximise → maximize

- colour → color

- modelling → modeling

- behaviour/neighbour → behavior/neighbor

- centred → centered

- travelling → traveling or if you are sticking to British English:

- -ization → -isation

- -ize → -ise

- Remember to hyphenate certain word pairs:

- ever changing → ever-changing

- real time → real-time

- second order → second-order

- start up → start-up

- steady state → steady-state

- state space → state-space

- two dimensional/three dimensional → two-dimensional/three-dimensional

- line 2 - In this paper a new... → In this paper, a new...

- line 3 - at low computational cost → at a low computational cost

- line 32 - question if → question of whether

- line 35 - This could possibly lead → this could lead

- line 43, 110, 112 - center line → centerline

- line 70, 128 - down stream → downstream

- line 72 - Changes of → Changes in

- line 73 - six-turbine-simulation → six-turbine simulation

- line 74 - large eddy simulation simulation → large eddy simulation

- line 79 - In this paper we... → In this paper, we...

- line 99 - Eventually a basic... → Eventually, a basic

- line 202 - assumes an uniform → assumes a uniform

- line 206 - wolrd → world (Reviewer 2: Line 206: wolrd should be world.)

- line 214 - at the cost calculating → at the cost of calculating

- line 222 - orientation remains the same → orientation remain the same

- Line 225: You write actuator disc theory, I would refer to this as 1D momentum theory, but I do understand that both terms can be used. (Reviewer 2)

- line 307 - to give an idea, how... → to give an idea how...

- Figure 8 (in the y-label) - differenece → difference

- line 340 - an dynamic → a dynamic

- line 351 - flow filed → flow field

- Figure 10 (caption) - as indicator → as an indicator

- line 373 - shows a similar behaviour → shows similar behavio(u)r

- line 376 - too fast → too-fast

- line 392 - Ghz → GHz

- line 392 - a SSD → an SSD

- line 396 - take away → takeaway

- line 416 - computational lightweight → computationally lightweight

- line 421 - shows a good performance → shows good performance

- line 427-428 - (repetition of 'improvement'. Consider rephrasing)

**Response**   Thank you for reading the text so thoroughly and finding these errors. All comments have been addressed.

**References**

[1] Larsen, G. C., Madsen, H. A., Thomsen, K., and Larsen, T. J. Wake meandering: a pragmatic approach. Wind Energy, 11(4):377–395, 2008.

[2] Madsen, H. A., Larsen, T. J., Pirrung, G. R., Li, A., and Zahle, F. Implementation of the blade element momentum model on a polar grid and its aeroelastic load impact. Wind Energy Science, 5(1):1–27, 2020.

[3] Pedersen, M. M., van der Laan, P., Friis-Møller, M., Rinker, J., and R´ethor´e., P. DTUWindEnergy/PyWake: PyWake, 2019.

[4] Stieren, A., Gadde, S. N., and Stevens, R. J. Modeling dynamic wind direction changes in large eddy simulations of wind farms. Renewable Energy, 170:1342–1352, 2021.